# Climate-driven invasion and incipient warnings of kelp ecosystem collapse

Scott D. Ling [1] ✉ & John P. Keane [1]

Climate change is progressively redistributing species towards the Earth's poles, indicating widespread potential for ecosystem collapse. Detecting early-warning-signals and enacting adaptation measures is therefore a key imperative for humanity. However, detecting early-warning signals has remained elusive and has focused on exceptionally high-frequency and/ or long-term time-series, which are generally unattainable for most ecosystems that are under-sampled and already impacted by warming. Here, we show that a catastrophic phase-shift in kelp ecosystems, caused by range-extension of an overgrazing sea urchin, also propagates poleward. Critically, we show that incipient spatial-pattern-formations of kelp overgrazing are detectable well-in-advance of collapse along temperate reefs in the ocean warming hotspot of south-eastern Australia. Demonstrating poleward progression of collapse over 15 years, these early-warning 'incipient barrens' are now widespread along 500 km of coast with projections indicating that half of all kelp beds within this range-extension region will collapse by ~2030. Overgrazing was positively associated with deep boulder-reefs, yet negatively associated with predatory lobsters and subordinate abalone competitors, which have both been intensively fished. Climate-driven collapse of ecosystems is occurring; however, by looking equatorward, space-for-time substitutions can enable practical detection of early-warning spatial-pattern-formations, allowing local climate adaptation measures to be enacted in advance.

Global climate change is causing widespread re-distribution of species and leading to cascading ecological change[1]. Concerningly, dire predictions of climate change warn of nonlinear "catastrophic" collapse with profound ecological, social and economic consequences[2]. Therefore, the capacity to learn quickly from early-warning signals of climate-driven collapse[3,4] is fundamental if such unwanted and irreversible changes are to be understood and ultimately avoided. The notion of acting 'before it is too late' is predicated on detecting warning signals of collapse well in advance, thus much theoretical and practical focus has attempted to define this holy grail[2–4].

Detection of early-warning signals has centered on the notion of 'critical slowing down' in high-frequency ecological time-series data typically obtained from spatially constrained systems using computationally intensive methods[3,4]. However, relatively few ecosystems are sampled frequently enough, over long enough, for such metrics to be practically useful[5]. In contrast, many ecosystems are experiencing accelerating and pervasive signatures of climate change impacts[1], which are occurring much faster than high-frequency long-term ecological baselines can be established from which early-warning metrics may be statistically defined[4].

For the threat of collapse posed by climate-driven species redistributions, early warnings may be signaled by detecting changes in spatial pattern formation[6] as warm-environment species advance their range poleward into historically cooler ecosystems across the globe[7]. In the marine environment of southeastern Australia, the East Australian Current (EAC) has strengthened over the past 60-years resulting in greater poleward penetration of warmwater and an approximate quadrupling of ocean warming rates compared to the global average[8].

[1]Institute for Marine & Antarctic Studies, University of Tasmania, Private Bag 129, Hobart, TAS 7001, Australia. ✉e-mail: Scott.Ling@utas.edu.au

Here, this pronounced change in the physical oceanography corroborates with increasing observations of poleward range-extensions of ~70 marine species from New South Wales (NSW) to eastern Tasmania in recent decades[9,10].

Of the species documented to have undergone range-extension to Tasmania, the warmwater Diadematid sea urchin *Centrostephanus rodgersii* (Agassiz) has extended its range south by ~700 km and is a most conspicuous and ecologically important arrival[11–13] due to its ability to destructively overgraze kelp forest habitat (Fig. 1a–d). The cascading impacts of overgrazing by *C. rodgersii* are profound, with a local loss of over 150 species, including highly lucrative commercial species such as abalone and rock lobster, that live amongst kelp habitat[11]. Within the central and southern NSW native range of *C. rodgersii*, this urchin, via its obligate scrapping mode of feeding on macroalgae including consuming holdfasts and juvenile macroalgal recruits, maintains barrens over more than 50% of all shallow reef[14] and similar levels of overgrazing are now apparent at some sites in northeast Tasmania. Sites most heavily impacted by overgrazing have warmed beyond the minimal thermal threshold for the urchins' development[15], with proximity of reefs to the EAC driving a steep latitudinal gradient in recruitment and population size dynamics southward along the Tasmanian eastcoast[12] (Fig. 1a). At local scales, field experiments comparing urchin survival inside versus outside Marine Protected Areas showed that urchin abundance is reduced in the presence of large predatory-capable spiny lobsters which have

been functionally extirpated across much of the east coast due to intensive harvesting[13].

This range-extending urchin was first detected in Tasmanian waters in 1960 and on the mainland northeast Tasmania at St. Helens in 1978 (Fig. 1a) with early-warning signs of overgrazing first apparent in eastern Tasmania during the early 2000s when small locally grazed patches within otherwise intact kelp forests were observed and termed 'incipient barrens'[16]. Incipient barrens represent a transient state progressing from small grazed patches (~1 m² scale) that become more numerous and coalesce to ultimately collapse kelp and form extensive barrens (10,000–1,000,000 m² scale) as urchin abundance locally increases (Fig. 1a–d). Based on a priori understanding of the urchin's destructive habit and the appearance of incipient barren spatial pattern formations[8], the range-extending urchin loomed as a major threat to the structure and functioning of Tasmanian reef systems[9]. Recognizing this threat was of critical importance as urchin overgrazing represents a "catastrophic phase-shift"[17], with collapse of productive kelp forests to low-value extensive urchin barren grounds ensured if range-extending urchin abundance exceeds the overgrazing tipping-point (approx. urchin biomass of 700 grams m⁻²; or ~≥2.2 urchins m⁻²). Contrastingly, recovery of kelp is very difficult once barrens have formed due to hysteresis, as the tipping-point for kelp recovery is almost a magnitude lower (approx. urchin biomass of 70 grams m⁻²; or ~≤0.36 urchins m⁻²)[17]. Urchin barrens, therefore, represent an alternative stable state that can be practically irreversible as urchins persist

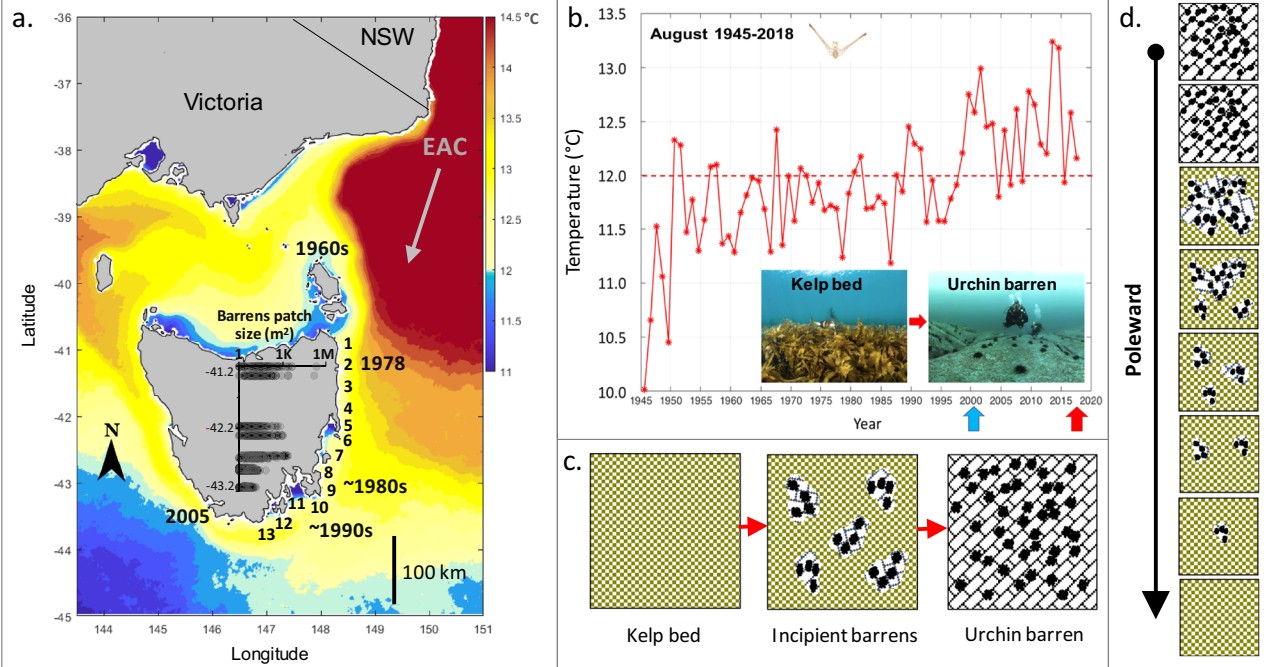

**Fig. 1 | Southeast Australian study system showing coastal warming, climate-driven invasion by the Longspined sea urchin, and spatial pattern formation of ecosystem collapse. a** Heatmap showing Sea Surface Temperature influence of the warm East Australian Current (EAC), which extends southward from New South Wales (NSW) to Tasmania where 13 east coast reef sites (listed 1–13 from north to south) were surveyed in 2001/02 and 2016/17 (map created in Matlab R2023a); years along the coast indicate the progressive poleward timeline of first occurrences of the sea urchin extending its range from NSW (SST data are means for August 2008–2018; IMOS multi-sensor SST composites: http://rs-data1-mel.csiro.au/thredds/catalog/imos-srs/sst/ghrsst/L3S-1d/catalog.html). Inset plot in **a** shows size-distribution of barrens patches (semi-transparent points plotted log-scale on x axis ranging 1, to 1 thousand, to 1 million m²), with progressively smaller barrens occurring southward along the invasion-front as estimated in situ by divers for a total of 1297 barrens patches during 2008 to 2011; source data are provided as a

Source Data file. **b** Long-term August (winter) sea temperature trend (1944–2017) from the Maria Island oceanographic station (Site 7): https://portal.aodn.org.au/); August is the month of spawning for the sea urchin *C. rodgersii*, with larvae (inset image) only developing above 12 °C (dashed horizontal line[15]), timing of surveys are shown as upward arrows on x axis (2001/02 = blue; 2016/17 = red); inset images taken in 2018 at Maria Island (Site 7, kelp) and St. Helens (Site 2, barrens). **c** Schematic of the spatial pattern of ecosystem collapse from kelp forests to urchin barrens via intermediate 'incipient barrens', which expand and coalesce as urchin density increases across reefs. **d** Schematic of observed climate-driven spatial pattern formation in the poleward direction; following from bottom-to-top provides a space-for-time substitution of the process of collapse from kelp to extensive barrens via appearance and ultimately coalescence of incipient barrens as the overgrazing tipping-point in urchin density is increasingly exceeded.

by switching diets from large kelps to microscopic and encrusting algae[17].

Here, we examine space-time dynamics of nonlinear ecosystem collapse over a 15-year period and examine the presence of incipient barren signals across the chronically warming coastline of eastern Tasmania (Fig. 1b–d). If early-warning signals are ultimately detectable, then knowing how to act, before it is too late, is contingent on understanding mitigating ecological circumstances. Thus, we also explore mechanisms of urchin invasion and overgrazing, respectively for different abiotic (i.e., reef substratum types/depth) and biotic covariates (i.e., predatory spiny lobsters plus competing native urchins and abalone abundances).

## Results

Across the region of range-extension (Sites 1–13), dive surveys to 18 m depth revealed the range-extending urchins' abundance to increase significantly from an average density of 0.10 (±0.05 SE; $n = 13$ sites) to 0.18 (±0.06 SE, $n = 13$ sites) individ m$^{-2}$ from 2001/02 to 2016/17 (Fig. 2a; Supplementary Table 1a; with site means calculated hierarchically from the mean of 3 subsites per site, with subsite means calculated from mean of 4 transects per subsite within each period). This x1.8 increase over 15 years equated to a compounding rate of population increase of 3.8% per annum. However, abundance of the range-extending urchin was uneven at the ~20 km site-scale (chiefly related to the proximity to the East Australian Current[8,12]), a feature which persisted through time (i.e., nil site-by-time interaction effect), and increase in abundance varied significantly at the local ~2 km subsite-scale (Supplementary Table 1a). For eastern Tasmania where overgrazing manifested (Sites 1–9), urchin abundance increased significantly from an average of 0.15 (±0.06 SE, $n = 9$ sites) to 0.26 (±0.07 SE, $n = 9$ sites) individ m$^{-2}$ (Fig. 2a); equating to an increase in abundance of x1.7 over 15 years. Contrastingly, for this section of coast, dive surveys revealed barrens to increase in cover by a factor of x3.9, from 2.3% (±1.6% SE, $n = 9$ sites) to 9.0% (±3.9% SE, $n = 9$ sites) cover (which also varied significantly at the subsite-scale, see Supplementary Table 1b), equating to a compounding rate of barrens increase of 9.5% pa (Fig. 2b).

The disproportionate collapse of kelp to barrens relative to increases in urchin density was revealed by an overall steepening in the negative trend between kelp cover and urchin abundance (Fig. 2c). Notably, the overgrazing collapse of kelp cover was counter to an overall increase in kelp cover outside of reef areas grazed by urchins, with kelp cover on a non-grazed reef in eastern Tasmania increasing from 2001/02 to 2016/17 due to compensatory thickening in cover of lower-lying warm-tolerant stipitate species (i.e., *Ecklonia radiata* & *Phyllospora comosa*) in the absence of surface-canopy giant kelp forests (formed by *Macrocystis pyrifera*) which declined (Supplementary Table 2). Divers also revealed a greater increase in urchin abundance on deeper reefs, particularly on large boulder reef >8 m depth (Fig. 3a–c; Supplementary Table 3a). Similarly, barrens also increased disproportionately in deeper water, with increases also greatest for large boulder reef (Fig. 3d–f; Supplementary Table 3b). Multiple regression showed a significant positive association between the range-extending sea urchin and native urchin (*Heliocidaris erythrogramma*), while both urchin species were negatively associated with predatory spiny lobsters (*Jasus edwardsii*) (Supplementary Table 4a). Abalone (*Haliotis rubra*) was negatively associated with urchin barrens (Supplementary Table 4b).

Extensive underwater towed-video covering 80 km in each survey period from 4 m to 40 m depth, revealed cover of early-warning 'incipient barrens' (categorized as three increasing patch-sizes) and fully collapsed 'continuous barrens' to all increase significantly through time across eastern Tasmania (Fig. 4a-d; see summary statistics Supplementary Table 5a–d). Greatest increase occurred for the smallest size-class of incipient barrens, i.e., the earliest forewarning of

collapse (Fig. 4a). Medium and large incipient barrens, as well as continuous barrens, increased consistently through time across sites (Fig. 4b, c), while small incipient barrens increased significantly at some sites but not at others where barrens had already expanded to larger patch sizes (Fig. 4d; Supplementary Table 5d). At the ~2 km subsite scale, increases in medium-to-large incipient barrens and continuous barrens varied significantly, yet evenness in the increase in small incipient barrens occurred at this scale (Supplementary Table 5a–d). Overall planar cover of barrens more than quadrupled from 3.4% to 15.2% (Fig. 4e), with significant variability in increase among subsites (Supplementary Table 5e). Increase in barrens equated to an expansion rate of 10.5% pa, with a projection of this unmitigated rate indicating barrens cover is poised to increase to ~50% in eastern Tasmania (Sites 1–9) as soon as the year 2029, consistent with that observed locally at St. Helens (Site 2) and as observed in NSW[14].

## Discussion

Nonlinear 'catastrophic' phase shifts can have profound socio-ecological consequences by collapsing ecosystem services and locking ecosystems into degraded undesirable states[2]. Our results reveal a population explosion of the range-extending urchin in Tasmania during an unprecedentedly warm 15-year period, with an even more abrupt increase in barrens, including a dramatic increase in the incipient warning signs of extensive collapse. This finding is highly consistent with the expectation of local exceedance of the overgrazing tipping point in urchin density[17]. Once this overgrazing tipping point is exceeded, kelp habitats appear to be 'living on borrowed time'[5]; which was evident for kelp forests at several sites in the early 2000s when many of the now overgrazed reefs occurred as largely intact kelp forests with incipient barrens and/ or urchins occurring beneath closed kelp canopies, as shown by the relatively flat trend between kelp cover *vs.* urchin density which steepened through time (Fig. 2c).

Critically, we show 'incipient barrens' to act as an effective early-warning signal prior to expansion and eventual catastrophic phase-shift to extensive urchin barrens. Prevalence of incipient barrens, which are currently present in some form across ~40% of reefs (Fig. 4a), clearly forewarns the potential for half of the eastern Tasmanian reefs to collapse to barrens within the coming decade. In 2001/02, the earliest warnings of overgrazing were manifest as small incipient barrens patches (1–10 s m$^2$) observed by divers to have formed on high-relief boulders, where protection from predators[18] and whiplash abrasion by large leathery kelps is afforded for grazing urchins[19,20]. During the study, such small incipient barrens were observed to increase in patch-size before coalescing with neighboring barrens patches to form barrens of >100 m$^2$ (S. Ling *pers. obs.*). Deep-to-shallow invasion and overgrazing were also indicated as urchins and barrens progressively encroached on shallower reefs to match shallower distributions observed within the urchins' native range[21]. Notably, in comparison to spatially restrictive and depth-limited dive surveys, towed video proved an efficient, scalable and cost-effective monitoring method well-suited to high-frequency spatially extensive sampling, which are key considerations for maximizing detection of spatial pattern formations indicative of impending ecosystem change.

Consistent with previously defined trophic-cascading effects of overfishing on urchin predators and kelp forest resilience[13,18], survey results showed range-extending and native urchins (*H. erythrogramma*) to be negatively associated with spiny lobsters (*J. edwardsii*). Abalone (*H. rubra*) was negatively associated with urchin barrens, as previously reported within NSW[22] and Tasmania[9]. While negative 'competitive exclusion' effects of abalone on urchins have been hypothesized, field experiments reveal negative effects of urchins on abalone but not of abalone on urchins[23]. As such, initial climate adaptation efforts have focused on increasing resistance to invasion and resilience of kelp bed habitats by rebuilding historically overfished populations of large predatory lobsters[13,18,24].

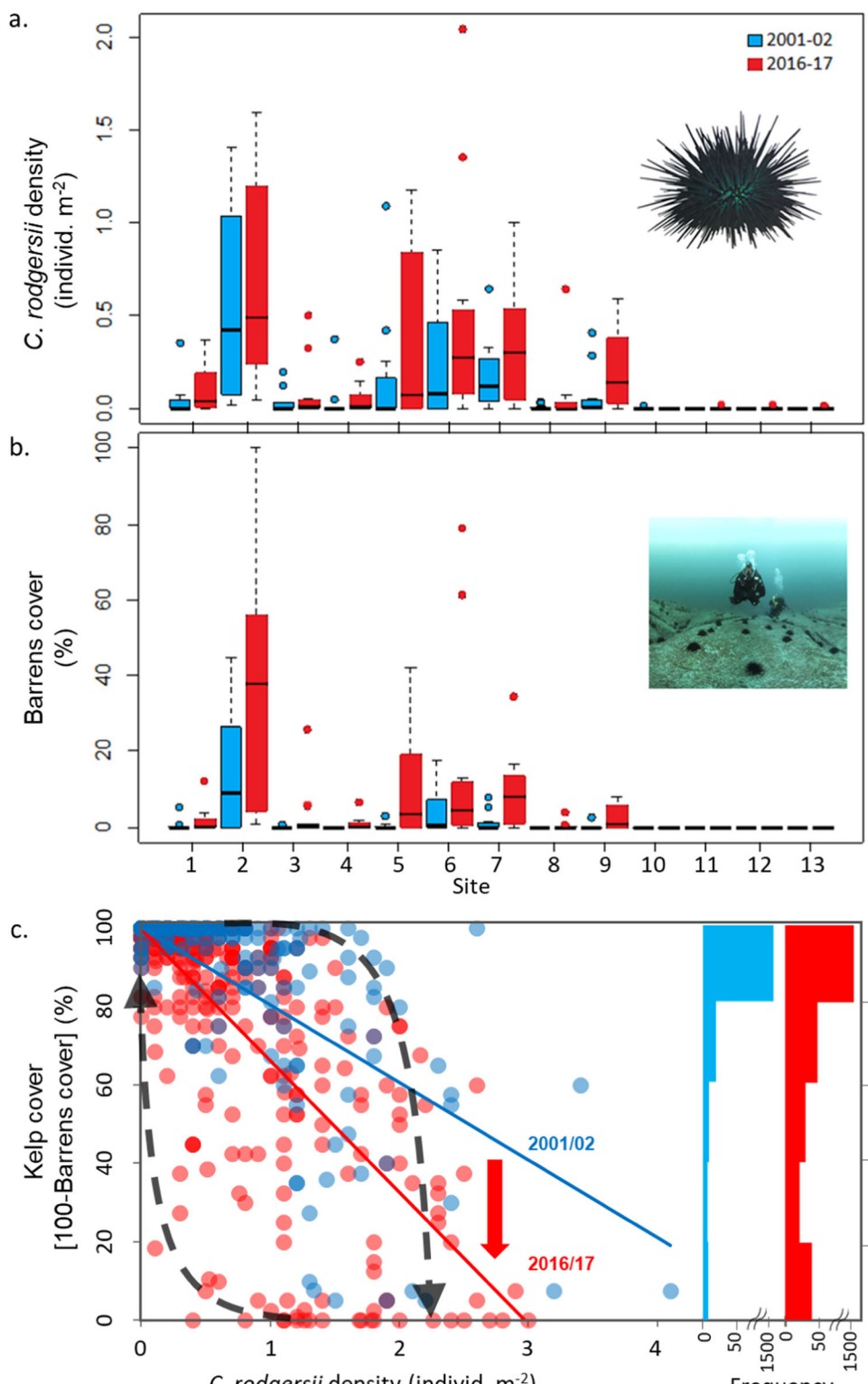

**Fig. 2 | Space-time dynamics of urchin invasion and catastrophic phase-shift from kelp to barrens.** Box plots of **a** *C. rodgersii* abundance, and **b** barrens coverage by period (blue = 2001–02; red = 2016–17) as assessed by divers, sites numbered north to south along coast (*n* = 3 subsites per site, with 4 transects within each subsite); box plots show median (50th percentile) as a horizontal bar, with upper and lower bounds of box defined as the 75th and 25th percentiles, whiskers extend to 1.5 times this interquartile range, outliers are shown as individual points occurring beyond >1.5 times interquartile range. **c** phase-shift dynamics between total kelp cover and *C. rodgersii* density through time as measured at 5 m² quadrat scale. Source data are provided as a Source Data file. **c** downward and upward dashed-arrows indicate tipping-points of overgrazing (i.e., -2.2 urchins.m⁻²) and kelp recovery (i.e., -0.36 individ.m⁻²)[17]; solid lines are simplified mean trendlines fitted to all data in 2001/02 (blue), $y = -19.82x + 100.31$, $R^2 = 0.56$, $n = 1600$ quadrats; and 2016/17 (red), $y = -33.58x + 99.85$, $R^2 = 0.65$, $n = 1600$ quadrats (red downward arrow indicates collapse of kelp through time); histogram on right of panel projects the frequency distribution of quadrats within 20% bins of kelp cover, showing increasing bimodality between high cover kelp beds and collapsed urchin barrens through time, 2001/02 (blue) to 2016/17 (red).

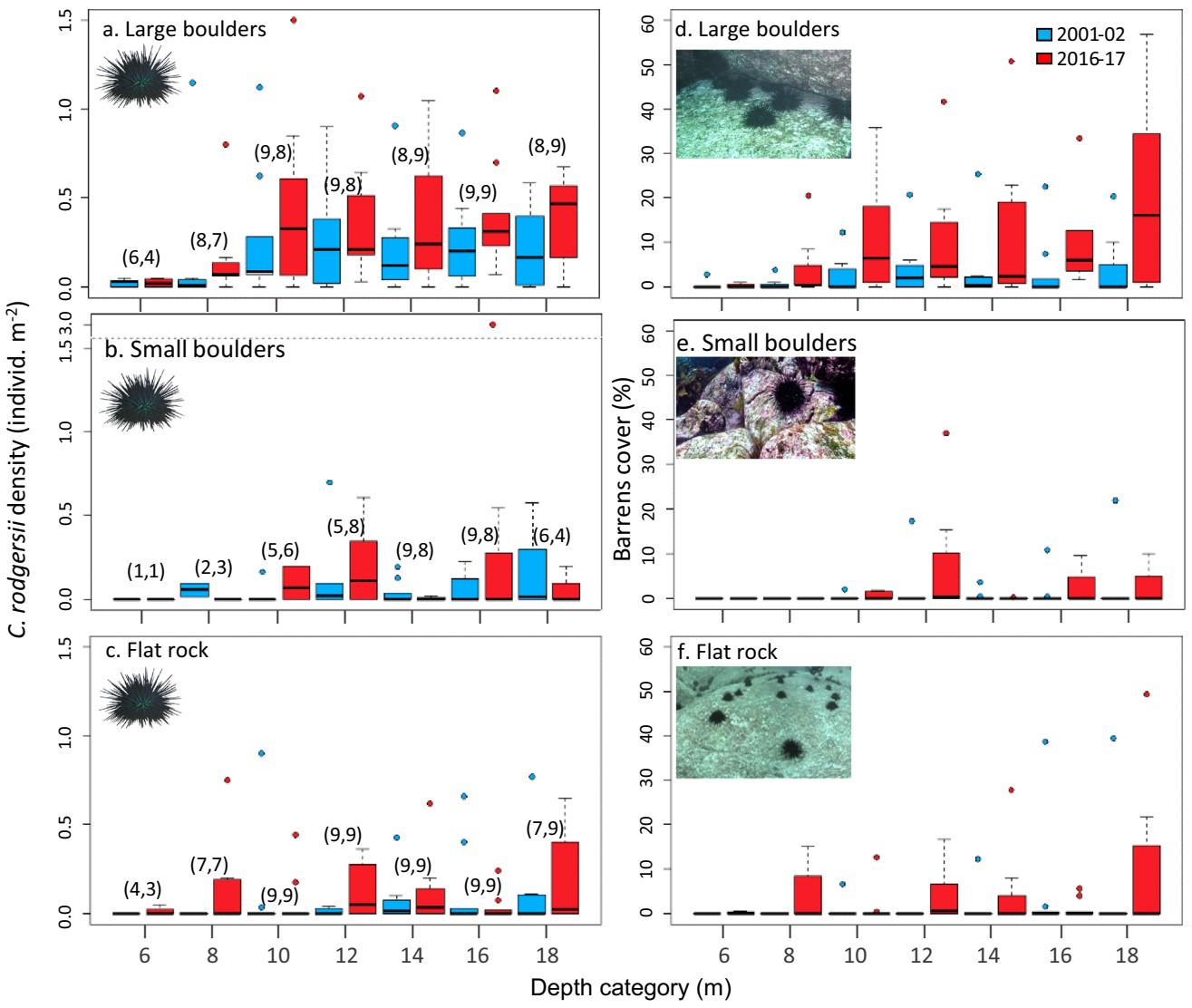

**Fig. 3 | Urchin invasion dynamics and overgrazed barrens by depth and reef type in eastern Tasmania.** Change in *C. rodgersii* abundance (**a**–**c**), and barrens cover (**d**–**f**) by substratum type and depth category (4–18 m depth) pooled for sites 1–9 in each period, box plots as per Fig. 2. Data are densities derived from 5 m² quadrats assessed by divers and averaged for quadrats dominated (>50% cover) by a particular substratum type within 2 m depth categories, depths shown represent the ceilings of each category; *n* quadrats within each category for both urchin abundance (**a**–**c**) and barrens cover (**d**–**f**) are shown in parentheses above boxes in (**a**–**c**) with each period separated by comma. Source data are provided as a Source Data file.

Importantly, while the early-warning signs are clear, most eastern Tasmanian reefs still currently exist as the desirable kelp state (~85% of reefs), thus, there is an opportunity to proactively heed the warning. Providing evidence for socio-ecological resilience and long-term reef ecosystem monitoring in combination with critical experiments identifying key mechanisms of overgrazing, coupled with strong science-industry connections, is enabling Tasmanians to combat the increasing threat of reef ecosystem collapse. Mitigation efforts underway include rebuilding historically overfished populations of large predatory lobsters under the "*East Coast Rock Lobster Stock Rebuilding Strategy*"[13,24], localised culling of range-extending urchins in key abalone fishing grounds[25,26], and a spatially variable catch-subsidy funded by the State Government to support the development of an urchin roe harvest industry that proactively directs harvest towards reefs at highest risk of overgrazing collapse[27]. Despite locally effective reductions of urchin densities below the 'overgrazing' tipping-point[25,26] and some reductions below the 'kelp recovery' tipping point (Scott Ling, *unpublished data*), recent surveys of two sites in southeast Tasmania during July 2021 indicate an exponential increase in population growth rate;

whereby increase in urchin density during the past 5-years is now equivalent to that observed over the previous 15 years. This tripling in population growth rate provides a dire warning and an urgent need to ramp-up mitigation efforts which are being heeded (22 tonnes of urchins were removed in a highly targeted and systematic government funded harvest over 5-days in 2023, J. Keane *unpub. data*).

Collective understanding of ecosystem collapse in space and time relies on rare examples monitored before and after abrupt change, as such, our study, along with other clear examples of climate-forced reef ecosystem collapse[28,29] provides an important bellwether of climate change impacts. Beyond previous studies, we reveal early warning of ecosystem collapse under climate change can be recognized well-in-advance by looking equatorward for signs of what to expect (i.e., looking in the equatorward direction to detect the emergence of novel spatial pattern formations, see Fig. 1d). That is, by effectively substituting space-for-time, opportunities for climate adaptation can be evidently identified and enacted before it is too late. This pragmatic outcome moves beyond intensive efforts to statistically define potential forewarnings of collapse from highly resolved time series

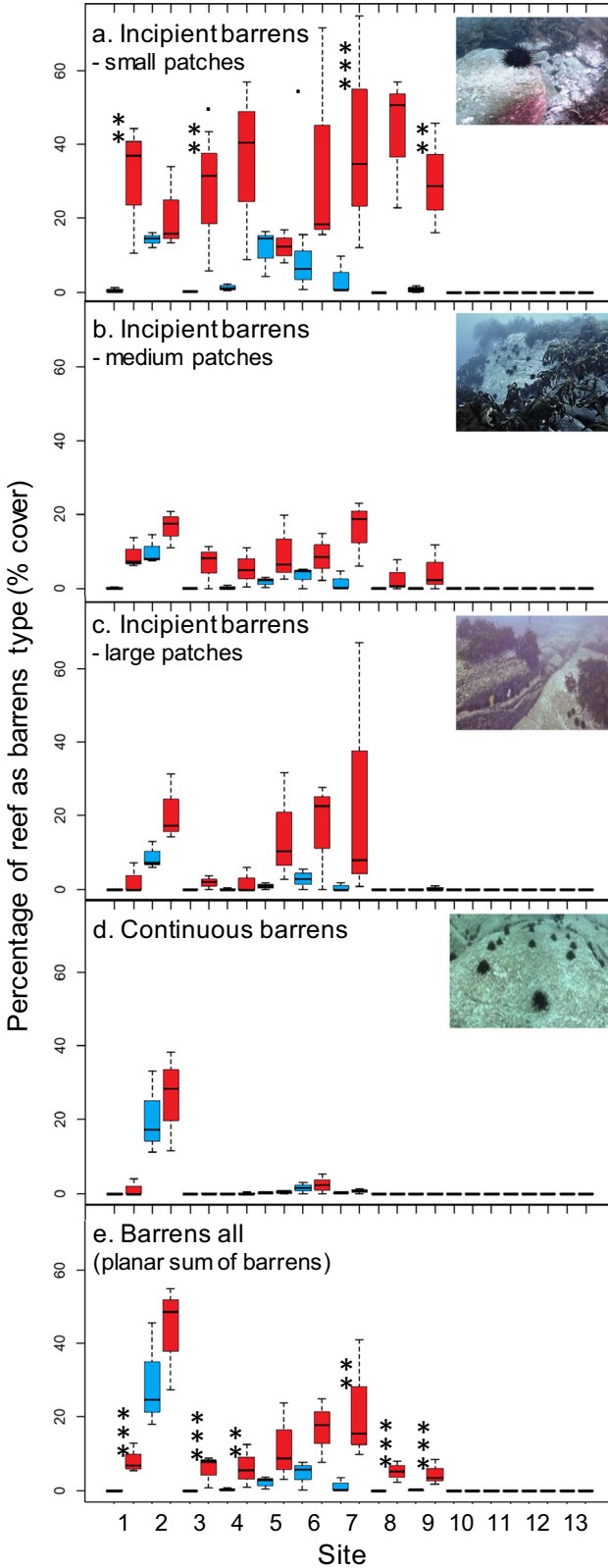

**Fig. 4 | Propagation of collapse from incipient to continuous barrens.** Incipient barrens of small [0.2–2 m diam.] (**a**), medium [2–4 m diam.] (**b**), and large [4–8.5 m diam.] (**c**) patch-sizes increased dramatically from 2001/02 (blue) to 2016/17 (red) indicating pending broadscale collapse to continuous barrens (**d**), with overall planar cover of barrens (**e**) also increasing markedly; the effect of "Time" was significant for all barrens response variables across eastern Tasmania, see Supplementary Table 5; box plots as per Fig. 2, with values at the site level generated from means of $n = 4$ video-tows ranging 4–40 m depth at $n = 3$ subsites. For incipient barrens (**a**) and planar sum of barrens (**e**), pre-planned one-tailed comparison of increase in percentage cover within each site is indicated by the significance codes: '**' <0.01, '***' <0.001. Source data are provided as a Source Data file.

practically impossible to know how to act as climate-driven invasions, and collapse of historically cooler ecosystems propagate in space and time.

## Methods

In 2016/17, scientific divers repeated the same methods used to survey 13 eastern Tasmanian sites (Fig. 1a) as originally surveyed during the 2001/02 'first assessment of the threat of *C. rodgersii*[16]. Sampling in both periods involved the consistent survey of a spatially hierarchical arrangement of 4 diver transects (spaced at ~200 m apart) at each of 3 subsites (spaced at ~2 km) within each of the 13 sites (spaced at ~20 km), giving 156 individual diver transects in total spanning 400 km from northeast to southern Tasmania. GPS was used to locate both the start and end positions of transects, which were set perpendicular to the shore, extending from ~4 m depth to a maximum of 18 m depth or a maximum total length of 100 m if the maximum seaward depth was shallower than 18 m. Based on reef topography across all sites, transects were ~50 m in length on average. On each transect line, a pair of buddy divers surveyed a 1 m swath each side of the transect line and for each 5 m section along the transect (giving a base contiguous quadrat size of 5 m²), recorded on slates the depth and abundance of urchins (*C. rodgersii* and *Heliocidaris erythrogramma*−noting that *H. erythrogramma* is not observed to form barrens on the open coast study region of eastern Tasmania[31]), southern rock lobster and blacklip abalone. Planar cover of substratum types within each contiguous 5 m by 1 m quadrat was estimated to the nearest 5%, i.e., resolved to a 0.5 by 0.5 m area, with substratum classified as either flat rock (>5 m effective diameter), large boulders (>1 m and <5 m diameter), small boulders (>0.2 m and <1 m diameter), cobble (>0.1 m and <0.20 m diameter), pebble (>0.01 m and <0.10 m diameter), gravel (<0.01 m diameter), or sand.

In addition to the temporal survey, a snapshot of barren patch-sizes across eastern Tasmania (see inset Fig. 1a) was estimated in situ by divers at sites 2, 5, 6, 7, 8, 9 from 2008 to 2011. A total of 76 geo-referenced timed swims (surface GPS towed by a diver for 30–45 min and ranging up to 550 m in length) spanning 5 to 15 m depth were conducted whereby divers searched for and scored barrens patch-sizes using a 1 × 1 m quadrat for calibration. The sizes of 1297 individual barrens patches were estimated in this way. Note that a barrens patch is an area of reef where all the kelp within the patch has been completely grazed away by urchins, resulting in bare-rock or encrusting/turfing filamentous algal forms (e.g., Fig. 4a, b insets), i.e., the percentage cover of kelp within a barrens patch is 0%, or alternatively 100% barren cover. As barrens patches become larger, urchin abundance within the barren also increases[24].

Percentage cover of urchin barrens was assessed by two methods: In situ SCUBA diver assessment along the same transects where urchin and invertebrate counts were recorded in depths from 4–18 m; and towed-underwater-video enabling sampling of barrens in depths >18 m and much greater spatial coverage. Diver-transect estimation of barrens cover, as per in situ assessment of substratum cover, involved each diver estimating the percentage cover of sea urchin barrens to the nearest 5% within 5 m by 1 m quadrats. Reef classified as urchin barrens

within spatially constrained systems. Furthermore, it recognizes the urgent need to 'look upstream' to combat the climate-challenges ahead, which will eventuate even under scenarios of immediate emissions reductions given committed global warming[30]. Irrevocably, in the absence of natural history understanding across biogeographical scales, it may not only be impossible to identify early-warning signs of pending ecological disasters, but it will also be

were characterized as intensively grazed by locally abundant urchins, which was discernible from reef lacking foliose macroalgae for other reasons, e.g., scour, and/ or dominance of sessile invertebrates.

Broader kilometer-scale patterns in barrens cover were estimated using a towed-underwater-video camera system. The spatially hierarchical sampling design was as per diver surveys, except that at the subsite level, two video transects were run perpendicular and two transects ran parallel with the shore. Perpendicular transects spanned 4–40 m depth (profiling available reef from the sand edge to shore), while parallel transects focused on the 15 m depth contour where *C. rodgersii* densities were observed to reach peak levels during original dive surveys. Parallel tows were ~1 km in length in straight-line distance, with the total surveyed distance of perpendicular and parallel video transects across all sites >80 km of reef in each survey period and the same start and end GPS of transects were used in each sampling period.

The video camera was towed ~1–2 m above the seafloor and/or algal canopy, which provided a swath width of ~3–4 m. Recorded video footage was time-stamped using onboard laptop computer dually capturing time, date, GPS position and depth via depth sounder. GPS information related to the boat, while the camera was offset on a towline ~20–30 m behind the boat which varied depending on depth and speed. Video footage was post-processed in the laboratory with barrens types classified for contiguous ~10 m intervals along each transect, which were summed and converted to percentages of the total transect length. Barrens were classified in four types: i.e., 'continuous' barrens whereby >85% barrens cover occurred within the camera field of view for 10 m, while the other three categories are different types of 'patchy' incipient barrens. 'Large' incipient barrens were defined as patchy barren where barrens covered >40% but <85% of the reef; 'Medium' incipient barrens defined patchy barrens in which barrens occupied between 20–40% cover; while 'Small' incipient barrens referred to patchy barren where barren cover was <20% cover. To obtain an overall planar estimate of barrens cover across all barrens categories, the proportion of each barrens type for each transect was multiplied by the mid-point of barrens cover as defined for that barrens type and then summed. That is, the percentage of 'continuous barrens' on a transect was multiplied by 0.925 (i.e., the mid-point between 85% and 100% barrens is 92.5%). Likewise, small, medium, and large barren categories were multiplied by their respective mid-points of 0.10 (i.e., mid-point between 1 and 20% cover was nominally 10%), 0.30 (mid-point between 20–40% cover was nominally 30%) and 0.625 (mid-point between >40–85% cover was nominally 62.5%) respectively.

Densities (no. individ. 5 m$^{-2}$) of sea urchins and cover of barrens and substratum types, plus depth, were averaged across neighboring 5 m by 1 m quadrats on either side of the transect line (i.e., by combining data from each buddy pair as assessed in 5 m by 1 m quadrats on either side of the transect line). To ensure densities and percentage covers were robust estimates per unit area of reef, abundances were converted to densities once the area of any sand in each diver quadrat or interval of video had been removed. To ensure temporal consistency in the depths sampled during each sampling period, diver and video transects were individually matched by depth (4–18 m depth) and transect length (i.e., in terms of the no. of 5 by 1 m quadrats per transect) between 2001/02 and 2016/17 sampling periods. Data were analyzed using analysis of variance (ANOVA) using a nested 2-factor mixed effects model testing the effects of "Time" (i.e., fixed effect, 2001/02 versus 2016/17) and "Site" (i.e., random effect of 13 eastern Tasmanian sites), plus the interactive term of "Time" by "Site", (i.e., variability in the "Time" effect across "Sites") and "Subsite" (3 per Site) nested within the "Time" by "Site" interaction term. The analysis of variance in *C. rodgersii* density and barrens cover at the Subsite-level (for both dive and towed-video estimates of barrens), within each time-period, was based on means of *n* = 4 transects. All statistical analyses were undertaken using R (R Development Core Team 2018), and

appropriate transformations of response variables were determined using the boxcox routine available in the MASS package in R.

Distribution of *C. rodgersii* density and barrens cover across depth, as assessed in situ by divers, was analyzed by defining 2 m depth strata: ranging 4–6 m, 6–8 m, 8–10 m, 10–12 m, 12–14 m, 14–16 m and 16–18 m. The influence of reef substratum type on *C. rodgersii* density and barrens cover was explored by assigning averaged neighboring quadrats to a dominant substratum type, i.e., the substratum type constituting >50% cover. Where substratum types were equally dominant, the dominant substratum was assigned as the smaller diameter category. Effects of depth and substratum were analyzed through time using a 3-way fixed effects ANOVA on data averaged across eastern Tasmanian sites 1–9 where both urchins and barrens were recorded on diver transects.

Relative contributions of time, latitude, substratum, depth plus other mobile benthic invertebrates including predatory lobsters, locally native sea urchins and abalone on *C. rodgersii* density and barrens was examined with multiple linear regression using *n* = 3200 quadrat-level estimates. Effects were partitioned using the "Relaimpo" package available in R (https://cran.r-roject.org/web/ packages/ relaimpo/relaimpo.pdf) using Lindeman, Merenda and Gold (LMG) estimation to determine the contribution of each explanatory variable to the overall $R^2$ of linear model fits. Correlations between explanatory variables of ≥0.70 or ≤−0.70 resulted in only one of the variables being retained, e.g., substratum types were highly correlated given their sum to 100% within quadrats, therefore only the "large boulder" substratum variable was retained.

### Reporting summary
Further information on research design is available in the Nature Portfolio Reporting Summary linked to this article.

## Data availability
Source data are provided with this paper, containing data from dive surveys of sea urchin barren sizes (sheet 1), sea urchin abundance, other invertebrates, substrate types including urchin barrens cover (sheet 2), plus towed-video surveys of urchin barrens (sheet 3). Oceanographic sea temperature data (Fig. 1) were sourced from Australia's Integrated Marine Observing System (IMOS)−IMOS is enabled by the National Collaborative Research Infrastructure Strategy (NCRIS), see https://mrs-data.csiro.au/thredds/catalog/imos-srs/sst/ghrsst/L3S-1d/ catalog.html.

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

## Acknowledgements

We acknowledge Craig Johnson for initiating the 2001/02 survey[16], funded by the Fisheries Research and Development Corporation (Project No. 2001/044). We thank a team of survey divers including Jeff Ross, Dave Stevenson, Robbie Kilpatrick, Simon Talbot, Sarah-Jane Pyke, Craig Sanderson, Gabrielle Dominguez, Edward Forbes, Jane Ruckert, Olivia Johnson, Laurel Johnston, Martin Filleul and Elizabeth Oh. Dave Stevenson, Jane Ruckert, Olivia Johnson, Ellie Paine and Amy Nau assisted with analysis of towed-video data. Ken Ridgway assisted with producing sea surface temperature plot in Fig. 1a. The 2016/17 survey was funded by the Sustainable Marine Research Collaboration Agreement, Tasmanian Government (to S.D.L.). This research was supported by the Australian Research Council (FT200100949 to S.D.L.).

## Author contributions

Conceptualization: S.D.L. Methodology: S.D.L. Investigation: S.D.L. (dive/video surveys in both periods), J.P.K. (dive/video surveys in 2016/17). Visualization: S.D.L. Funding acquisition: S.D.L. Project administration: S.D.L. 2001/02 survey; J.P.K. 2016/17 survey. Writing—original draft: S.D.L. Writing—review & editing: S.D.L., J.P.K.

## Competing interests

The authors declare no competing interests.
