## [Peer Review File · Nature Communications]

Climate-driven invasion and incipient warnings of kelp ecosystem collapseREVIEWER COMMENTS

Reviewer #1 (Remarks to the Author):

This is a very important paper with significance that extends well beyond SE Australia. Collapses of complex ecosystems often surprise managers and the scientists studying them. Ling and Keane effectively apply contemporary theories on ecosystem forecasting to document conditions leading to, and foretelling, the collapse of Tasmanian kelp forest ecosystems. The authors describe a trophic cascade, in which fisheries induced declines of large rock lobster predators and warming ocean temperatures facilitate the expansion of sea urchins that graze down kelp forests. The resulting "barren" state has low productivity, is species depauperate and has reduced commercial value.

The Ling and Keane study illustrates well some of the warning signs of collapse for complex marine ecosystems. This is the first study I've seen that documents changes in in kelp-deforested "barren" patches in both space and time. While our changing climate is unmanageable at decadal time scales, some of the human-induced changes to local marine food webs can be reversed relatively rapidly and thus prevent incipient barren patches from becoming sea urchin barren stable states.

The authors describe a viable plan for identifying and reversing the kelp forest ecosystem from becoming a barren state. Sea urchin gonads (roe) are a highly valued food so harvesting sea urchins would be a practical solution. However, this is most viable where barren patches are surrounded by kelp which they define as "incipient barrens" (e.g., Fig. 1C). The importance of taking management actions during this early warning phase is that once kelp forests are entirely overgrazed, sea urchins do not die but they reabsorb their gonads and thus lose their economic value for sea urchin harvesting.

This study is relevant globally as kelp forests are declining world-wide (Filbee-Dexter and Scheibling 2014). Such declines often result from the loss of sea urchin predators (Steneck et al 2004, Norderhaug et al 2014). Ling and Keane suggest that fisheries management can improve kelp forest ecosystems and maintain a desirable stable state. There is evidence for this thesis in the Northern Hemisphere where the combination of sea urchin harvesting and

invertebrate predators released from fishing pressure maintained the preferred kelp forest alternative stable state (Steneck et al 2013).

The authors point out that warming seas not only improve conditions for sea urchin larvae, but also facilitate changes in the species composition of algal communities. This too has been observed globally with species of foliose macroalgae replacing kelp species. This creates ecosystems with greater carrying capacity and ecosystem services than the crustose coralline dominated "sea urchin barren" state.

Readers should understand that the authors do not think the climate-driven aspect of this paper can be managed directly. This study is on the indirect effects of climate change. So, it differs from the well documented decline of coral reefs following acute warming events. There is no way to manage or mitigate ocean warming impacts on coral reefs. However, kelp forest ecosystems are different because it is possible to identify "incipient states" of change for management actions to be applied at an ecosystem scale with relatively rapid results.

Literature cited

Filbee-Dexter, K. and Scheibling, R.E., 2014. Sea urchin barrens as alternative stable states of collapsed kelp ecosystems. *Marine ecology progress series*, 495, pp.1-25.

Norderhaug, K.M., Christie, H., Pedersen, M.F. and Fredriksen, S., 2014. Predators of the destructive sea urchin *Strongylocentrotus droebachiensis* on the Norwegian coast. *Marine Ecology Progress Series*, 502, pp.207-218.

Steneck, R.S., Graham, M.H., Bourque, B.J., Corbett, D., Erlandson, J.M., Estes, J.A. and Tegner, M.J., 2002. Kelp forest ecosystems: biodiversity, stability, resilience and future. *Environmental conservation*, 29(4), pp.436-459.

Steneck, R.S., Leland, A., McNaught, D.C. and Vavrinec, J., 2013. Ecosystem flips, locks, and feedbacks: the lasting effects of fisheries on Maine's kelp forest ecosystem. *Bulletin of Marine Science*, 89(1), pp.31-55.

Reviewer #2 (Remarks to the Author):

This is well-written and interesting manuscript that explores the poleward extension of a sea urchin grazer and resulting incipient barrens as an early warning sign of climate change-driven ecosystem collapse. I found the manuscript technically sound and the results compelling, although I provide a few considerations for the authors below. Overall, I believe this research will be of interest to the field, but its broader significance may be limited only in the sense that urchin barrens are well-documented as a driver of ecosystem collapse, and they have been repeatedly linked to climate change-driven perturbations such as marine heatwaves, declines in macroalgae, and loss of predators. However, it is particularly interesting that the incipient barrens described in this study are the result of the range extension of a dominant reef grazer. My main comments are therefore to consider the generality of incipient barrens as early warning signs and whether the warning is specific to the extension of a grazer into novel habitats. I provide several thoughts on this below.

Major comments:

-The ms suggests that incipient barrens have the potential to deforest half of eastern Tasmanian reefs (like 182) within coming decades. While this is certainly a possibility, sea urchin barrens have been documented around the world at the spatial magnitude of meters to hundreds of kilometers (reviewed in Filbee-Dexter and Scheibling 2014). Some persist for only a season and others have persisted for many decades. In my opinion, understanding the environmental and ecological contexts that explain the likelihood of incipient barrens persisting and expanding is the fundamental link needed to explain when and why incipient barrens are an early warning sign.

-It is interesting that in this system ecosystem collapse was due to the poleward expansion of a particular grazer species (*C. rodgersii*) into novel habitat. Perhaps the early warning sign is not just the emergence of incipient barrens, but specifically incipient barrens formed by a particular species in novel habitat. If this is correct, please emphasize this in the manuscript. It is not just range extensions or redistribution of grazers with specific thermal affinities, but barrens have been documented globally and attributed to native grazers as well. The authors mention the inclusion of the native sea urchin as a covariate, but if it is indeed the

case that these barrens are driven by the extension of *C. rodgersii* and not the native species, then this should be emphasized. This could be supported with the addition of density of the native species in Figure 2C or as a supplementary figure.

-Unless I am missing something, why not consider the total absolute density of both the native urchin and also *C. rodgersii*? It looks like they are treated independent in Table SI5. It would be interesting to see a plot of the baseline and temporal densities of both species over time.

Line 122 – I recommend removing ‘resistance and resilience’ unless the authors provide some specific definitions for these terms as context, along with carefully defined statistical tests. Briefly, I interpret resistance and resilience as ecosystem-level responses where resistance is the capacity of a system to maintain community structure (or functioning, depending on goals) despite a perturbation, and resilience is both resistance to- and recovery from-perturbation (*sensu* Bates et al. 2019). Perhaps there is a different interpretation in this system, but please provide clear definitions of these terms if they are retained.

The ms uses the terms ‘phase-shift’ and ‘stable state’ in a few different places. It seems that ‘phase-shift’ is used to describe the changes here, which I think is correct, but as a reader it would be helpful to see this defined. Dudgeon et al. 2010 define a stable state as a novel configuration that persists after the environmental conditions return to what they were before a perturbation, which is not the same of a phase shift. I think the terms are used correctly here, but line 109 uses “phase shift” and 115 uses “alternative states.”

Perhaps I missed this, but I don’t see how the critical ‘tipping points’ were derived other than an approximation from Figure 2C. I looked up reference 17 but I don’t see a mention of the tipping points included here. The authors have the data to quantitatively derive these tipping points. The way I have seen this done (and recommend) is that a discriminant analysis is first used to assign each replicate (quadrat or site) a binary value (1 = barren, 0 = forest) based on several predictors (substrate type, algal cover, depth, etc.) and this becomes the response for a logistic regression. The logistic regression is then performed

using urchin density as the predictor and the tipping point thresholds are the density values at which it becomes likely (50%, 80%, 95%, whatever probability you choose) that the system will shift to barren or return to forest. The challenge with the recovery arrow is it depends on replicates where barrens were observed transitioning to forests. Without that, it is just a single tipping point based on correlative observations between kelp density and urchin density. Please consider this type of a model to identify the tipping points and whether there are sufficient 'reverse-shift' replicates to identify the reverse-shift threshold.

Minor comments:

I am curious about kelp cover over time. Without a plot of the time series of kelp cover it is difficult to assess whether increases in urchin density proceed kelp loss or just amplify macroalgae declines that are the result of already poor environmental conditions. I am not sure whether these data are available since the study used two discrete time periods, but in order to really attribute barren expansion as a driver of kelp loss (which completely makes sense), it requires a temporal component. Otherwise it is correlative and kelp loss could have preceded increases in urchin density, or led to emergence from refuge.

Line 69-72 – this is really interesting that *C. rodgersii* are so prevalent!

Line 77 – do lobsters prey on *C. rodgersii* specifically, or would this be a novel species interaction?

Line 127 – It is not just the effect of absolute abundance but also change in grazer behavior. Some highly forested areas can maintain high densities of cryptic passively-grazing sea urchins, so it is the behavioral emergence from refuge resulting from resource availability (decline in macroalgae due to warming) that can feedback to influence declines (Smith & Tinker 2022, Karatayev et al. 2021). In Figure 2A it looks like the mid latitude sites had historically high densities relative to the north and south (with the exception of St. Helens) and also the relatively highest densities after 2016. Were these deep sites with large boulders? It looks here that the pattern is the same, just amplified. I.e., sites with historically high densities of urchins continued to have the relatively highest densities.

Lines 140-142 – the difference in slopes seems to be heavily influenced by two or three quadrats in the 01/02 period with high kelp cover despite high urchin densities (blue points, Figure 2C), but there is clearly less kelp at lower urchin densities in the 2016/17 period. It would be worthwhile to at least compare the slopes after remove those two or three extreme quadrats.

Line 289-129 – please parenthetically include the error associated with these means.

Figures:

Figure 1. Love this figure! However “Barrens patch size” is difficult to read. Please make the text black.

Figure 2. In a) there was clearly an increase in the mean density between periods, but there was an even more pronounced increase in the range across replicate quadrats with a skewness towards a wider density range above the mean. This is an interesting result that could suggest underlying patch-level dynamics. In a and b) please increase the transparency of the filled area and show the datapoints. c) the reverse shift arrow is difficult to see. Please change the color or make it darker. Also, I know that the tipping point arrows are approximate, but where these approximated by visual inspection or using a quantitative approach? Unless I am missing something, I don't see how the 2.2 and 0.36 thresholds were determined. Please see my comment about the discriminant analysis and consider whether the recovery arrow is appropriate depending on the number of replicates where true recovery was observed. If recovery was not observed, then hysteresis is speculative and the relationship between kelp cover and urchin density should be represented by a single line or curve (single tipping point for forward and reverse shifts).

Figure 3. Please add the datapoints and make the filled area more transparent so that we can see them. I would reorder from bottom to top: flat rock, small boulders, large boulders.

Figure 4. Please add text indicating the patch size to each facet to remind us the extent of

'small' 'medium' and 'large'.

References in response:

Filbee-Dexter, K., & Scheibling, R. E. (2014). Sea urchin barrens as alternative stable states of collapsed kelp ecosystems. *Marine ecology progress series*, 495, 1-25.

Karatayev, V. A., Baskett, M. L., Kushner, D. J., Shears, N. T., Caselle, J. E., & Boettiger, C. (2021). Grazer behaviour can regulate large-scale patterning of community states. *Ecology Letters*, 24(9), 1917-1929.

Smith, J. G., & Tinker, M. T. (2022). Alternations in the foraging behaviour of a primary consumer drive patch transition dynamics in a temperate rocky reef ecosystem. *Ecology Letters*, 25(8), 1827-1838.

Bates, A. E., Cooke, R. S., Duncan, M. I., Edgar, G. J., Bruno, J. F., Benedetti-Cecchi, L., ... & Stuart-Smith, R. D. (2019). Climate resilience in marine protected areas and the 'Protection Paradox'. *Biological Conservation*, 236, 305-314.

Reviewer #3 (Remarks to the Author):

Summary:

Shifting environmental conditions are redistributing species across the globe. This study focuses on a range expansion of a sea urchin native to Australia to kelp forests off the coast of eastern Tasmania. Replicate surveys conducted across 13 sites and 15 years apart are described. These include diver surveys of urchin, lobster, and abalone abundance, and percent cover of substrate, including barren. Additionally, the authors conducted similar video surveys of planar cover of total urchin barrens and of different barren sizes in order to show the evolution of small incipient barrens (a starting condition) to large and continuous barrens that represent a state shift in the system. The authors found that incipient barrens increased through time, especially in northeastern and eastern Tasmania (southeastern

Tasmania showed few urchins and barrens). The authors also showed how increases in the abundance in urchins was associated with disproportionately increased barren area. Deeper areas of reef with large boulders were associated with greater urchin abundance and there were negative associations between urchin abundance and the predatory lobster. The authors conclude with a discussion of using these early warning signals to initiate further research, partnerships, and ecosystem defense strategies.

General comments:

Overall, I thought this was a well written and timely study worthy of publication. The two most important results from the study, in my opinion, are the data showing the temporal progression of urchin abundance and barren cover/size across space and the clear change in slope between barren area and urchin density between the two time points, implying a non-linear relationship. I did have some comments on how to improve the manuscript and reproducibility.

First, I have some trouble with the authors stating that barren cover may increase to 50% by 2029. I understand that this is an extrapolation on the measured barren expansion rate between the two survey dates. I also feel that the authors intuition is likely correct, especially considering that there are multiple stressors that are all trending towards a more hostile environment for kelp. However, I am not sure the data support this conclusion when viewed in a different way. First, we start with looking at sites 1 – 9, defined as eastern Tasmania by the authors. From the planar barren data in Supp Data 3, the site with a highest urchin barren area in the 2001-2002 surveys (Site 2; St Helens) had a mean urchin barren area of ~23%. In the later surveys in 2016-2017, it had a mean urchin barren area of ~42.5%, representing an increase of just under 20% in 15 years. All of the other sites had vastly lower barren areas in the 2001-2002 surveys, and sites 1, 3, 4, 8, and 9 all has less than 10% barren area in the 2016-2017 surveys (with site 5 having just over 10% and site 6 and 7 having about 17.5%). It could be that there are non-linearities in the rate of barren expansion that are not being considered when observing the early stages of barren formation (i.e., is there a greater rate of barren expansion when moving from 0% to 10% than from 20% to 50%) at sites using only two time points. While I personally believe the authors may be correct (due

to both increases in urchin abundance and temperature in this system, along with personal experience observing urchin barrens in other areas of the world), I would caution the authors to word these statements carefully. Additionally, while the authors do state that eastern Tasmania only covers sites 1 – 9 (and not the southern sites 10 – 13 where little barren cover has been observed in the data presented), there is a chance that people will interpret these findings as the entire eastern coast of Tasmania. Another reason to be as clear as possible when stating these predictions in the manuscript.

My other concern is with the data files that accompany the manuscript. I found a few discrepancies (or edits) that need to be addressed before publication. First, in Figure 2, n is defined as 1730 for the 2001-2002 surveys and 1727 for the 2016-2017 surveys, but the data file contains 1600 for both. I see the data files says (in the sheet name) that 'no sand' has been truncated. This may lead to the discrepancy and must be addressed. I found the stated statistics in the figure to be nearly identical. Second, using the planar urchin barren cover I found sites 1 – 9 to have 3.22% barren cover in the 2001-2002 surveys and 12.72% in the 2016-2017 surveys. Similar to what is stated in the paper but different (it may also lead to a slightly different rate of change for your extrapolation). Maybe I am looking at the wrong data, but others may do the same. Third, I was a bit confused on the proportion of barren sizes in the planar barren area data sheet. I think these data need more explanation in the manuscript so that readers can analyze these data correctly. These are the three that were most apparent in my analysis of the data and there may be others, so I think a careful reanalysis of the data provided is warranted.

Specific comments:

Define the abbreviation for New South Wales (NSW) in paragraph three of the introduction.

Misspelling of 'survey' in the acknowledgments.

Reviewer #4 (Remarks to the Author):

This paper shows an extensive amount of kelp and sea urchin data collected in Tasmania, comparing 2001/2002 and 2016/2017. The authors stress the current increases in urchin barren cover in warming coastal zones (poleward). Besides, they focus on the potential for early-warnings of a catastrophic shift towards complete urchin barrens, by monitoring incipient spatial patterns.

As a theoretical ecologist, I'm not familiar with the field methods used, so I won't be able to comment on these. I will focus on the link to the theory made by the authors, and the interpretation on the data in the context of alternative stable states and early warning. First of all, I'm impressed by the richness of the data. I'm not completely convinced of the entire line of reasoning, and the support by the data. I would like to share my thoughts on this, and some suggestions below.

Firstly, I miss a discussion on the scale of the theoretical dynamics and the data. Are the observed patterns assumed to be stable, or unstable transient states where small incipient patches with high numbers of urchins slowly expand into the kelp? In the last case, is the early warning not actually the start of the (maybe slow) collapse? Or in the first case, how catastrophic is the collapse then? A reference is made to Rietkerk et al, but any discussion on the scale of the described feedbacks, the scale of the tipping points, and the scale of the data is lacking.

One of the first points made in the paper, is that the kelp-sea urchin system has alternative stable states, and shows catastrophic shifts at a certain level of urchin cover. This data seems to support this, although it is not completely clear to me at what scale (see first point) (if I understand correctly 50m). To illustrate the hypothesized existence of alternative stable states, it would be insightful to illustrate the frequency distribution vertically next to Fig. 2c, to show bimodality in the data.

The authors draw two arrows in Figure 2c indicating tipping points. I understand that these levels are based on earlier research, but since they have no relation to the data studied itself, the drawn trajectories give the reader a bit of a false idea of a directional change. I would suggest to draw actual observed trajectories by drawing arrows for each of the sites between the two observation periods.

It is unclear to me how Figure 3 relates to the main point of the paper. Maybe I miss something, and could it just be stressed a bit more.

How the data is presented at the moment, I don't see that mapping patches and patch sizes would help for early warning. How do the patch sizes compare to urchin barren cover? Can results from Figure 2 and patch sizes as measured in Figure 4 be compared? Can actually urchin barren cover be used as an early warning sign, where simply the occurrence of patches is the first warning (so from 0 to >0% cover), or is there more information in the patch sizes measured? For instance, does cover stay the same, but patches become larger at sites where there are more urchins? This could easily be shown in a plot, and helps the reader to convince the usefulness of looking at patches and patch sizes in particular. Overall, I think the link between theory and data can be strengthened more to convince the reader. Also, the bridge to actual early warning signs (cover? patches?) needs some consideration. Furthermore, I think with some adaptations, it can become a rich paper, discussing an important question, whether (the increase of / occurrence of) incipient patches can be used as warning signs of systemic kelp collapse.

REVIEWER COMMENTS

Reviewer #1 (Remarks to the Author):

This is a very important paper with significance that extends well beyond SE Australia. Collapses of complex ecosystems often surprise managers and the scientists studying them. Ling and Keane effectively apply contemporary theories on ecosystem forecasting to document conditions leading to, and foretelling, the collapse of Tasmanian kelp forest ecosystems. The authors describe a trophic cascade, in which fisheries induced declines of large rock lobster predators and warming ocean temperatures facilitate the expansion of sea urchins that graze down kelp forests. The resulting "barren" state has low productivity, is species depauperate and has reduced commercial value.

The Ling and Keane study illustrates well some of the warning signs of collapse for complex marine ecosystems. This is the first study I've seen that documents changes in in kelp-deforested "barren" patches in both space and time. While our changing climate is unmanageable at decadal time scales, some of the human-induced changes to local marine food webs can be reversed relatively rapidly and thus prevent incipient barren patches from becoming sea urchin barren stable states.

The authors describe a viable plan for identifying and reversing the kelp forest ecosystem from becoming a barren state. Sea urchin gonads (roe) are a highly valued food so harvesting sea urchins would be a practical solution. However, this is most viable where barren patches are surrounded by kelp which they define as "incipient barrens" (e.g., Fig. 1C). The importance of taking management actions during this early warning phase is that once kelp forests are entirely overgrazed, sea urchins do not die but they reabsorb their gonads and thus lose their economic value for sea urchin harvesting.

This study is relevant globally as kelp forests are declining world-wide (Filbee-Dexter and Scheibling 2014). Such declines often result from the loss of sea urchin predators (Steneck et al 2004, Norderhaug et al 2014). Ling and Keane suggest that fisheries management can improve kelp forest ecosystems and maintain a desirable stable state. There is evidence for this thesis in the Northern Hemisphere where the combination of sea urchin harvesting and invertebrate predators released from fishing pressure maintained the preferred kelp forest alternative stable state (Steneck et al 2013).

The authors point out that warming seas not only improve conditions for sea urchin larvae, but also facilitate changes in the species composition of algal communities. This too has been observed globally with species of foliose macroalgae replacing kelp species. This creates ecosystems with greater carrying capacity and ecosystem services than the crustose coralline dominated "sea urchin barren" state.

Readers should understand that the authors do not think the climate-driven aspect of this paper can be managed directly. This study is on the indirect effects of climate change. So, it differs from the well documented decline of coral reefs following acute warming events. There is no way to manage or mitigate ocean warming impacts on coral reefs. However, kelp forest ecosystems are different because it is possible to identify "incipient states" of change for management actions to be applied at an ecosystem scale with relatively rapid results.

Literature cited

Filbee-Dexter, K. and Scheibling, R.E., 2014. Sea urchin barrens as alternative stable states of collapsed kelp ecosystems. *Marine ecology progress series*, 495, pp.1-25.

Norderhaug, K.M., Christie, H., Pedersen, M.F. and Fredriksen, S., 2014. Predators of the destructive sea urchin *Strongylocentrotus droebachiensis* on the Norwegian coast. *Marine Ecology Progress Series*, 502, pp.207-218.

Steneck, R.S., Graham, M.H., Bourque, B.J., Corbett, D., Erlandson, J.M., Estes, J.A. and Tegner, M.J., 2002. Kelp forest ecosystems: biodiversity, stability, resilience and future. *Environmental conservation*, 29(4), pp.436-459.

Steneck, R.S., Leland, A., McNaught, D.C. and Vavrinc, J., 2013. Ecosystem flips, locks, and feedbacks: the lasting effects of fisheries on Maine's kelp forest ecosystem. *Bulletin of Marine Science*, 89(1), pp.31-55.

Author reply:

Reviewer #1 recognises the importance of this research and strongly endorses the manuscript. In the final paragraph of comments, we note that Reviewer #1 highlights that our manuscript focusses on local actions that can be taken to reduce the risk of climate change. While we think it is important to act locally, we also think it is critical to take global actions to reduce emissions. In our revised version we have added text and a reference in the final paragraph to emphasise the synergy between global and local climate change action.

The second last sentence of the manuscript now reads: "Furthermore, it recognises the urgent need to 'look upstream' to combat the climate-challenges ahead, which will eventuate even under scenarios of immediate emissions reductions given committed global warming³².

- 32 IPCC, 2021: Summary for Policymakers. In: *Climate Change 2021: The Physical Science Basis. Contribution of Working Group I to the Sixth Assessment Report of the Intergovernmental Panel on Climate Change* [Masson-Delmotte, V. P., et al. (eds.)]. Cambridge University Press, Cambridge, United Kingdom and New York, NY, USA, pp. 3–32, doi:10.1017/9781009157896.001.

Reviewer #2 (Remarks to the Author):

This is well-written and interesting manuscript that explores the poleward extension of a sea urchin grazer and resulting incipient barrens as an early warning sign of climate change-driven ecosystem collapse. I found the manuscript technically sound and the results compelling, although I provide a few considerations for the authors below. Overall, I believe this research will be of interest to the field, but its broader significance may be limited only in the sense that urchin barrens are well-documented as a driver of ecosystem collapse, and they have been repeatedly linked to climate change-driven perturbations such as marine heatwaves, declines in macroalgae, and loss of predators. However, it is particularly interesting that the incipient barrens described in this study are the result of the range extension of a dominant reef grazer. My main comments are therefore to consider the generality of incipient barrens as early warning signs and whether the warning is specific to the extension of a grazer into novel habitats. I provide several thoughts on this below.

Author reply:

Reviewer #2 identifies that the capacity of urchins to drive ecosystem collapse which is indeed well established in the literature and which I reviewed and compiled global pattern in 2015 (Reference #4 - Ling et al. Global regime shift dynamics of catastrophic sea urchin overgrazing. *Philosophical Transactions of the Royal Society B: Biological Sciences* 370, 20130269 (2015). However, the spatial expression of pending ecosystem collapse has not been reported for urchins or indeed rarely for other ecological systems and none as an indicator of a climate change related collapse. In our example, the appearance of incipient barrens and progression to large scale collapse is the critical finding of the study and forewarning of extensive ecosystem collapse driven by climate change. The generality of our findings is that change in patchiness of habitats can be a very real indicator of climate driven impacts and that identification and monitoring of incipient spatial features can provide an early-warning system of collapse.

Major comments:

-The ms suggests that incipient barrens have the potential to deforest half of eastern Tasmanian reefs (like 182) within coming decades. While this is certainly a possibility, sea urchin barrens have been documented around the world at the spatial magnitude of meters to hundreds of kilometers (reviewed in Filbee-Dexter and Scheibling 2014). Some persist for only a season and others have persisted for many decades. In my opinion, understanding the environmental and ecological contexts that explain the likelihood of incipient barrens persisting and expanding is the fundamental link needed to explain when and why incipient barrens are an early warning sign.

Author reply:

We agree that urchin barrens vary on a range of sub-metres to 10s of kilometre scales around the world. They also vary at a range of temporal scales. Smaller barrens are much more labile, expanding or shrinking/ disappearing, compared to larger barrens patches. Across multiple scales, spatial and temporal variation in barrens patches does, in large part, relate to environmental and ecological contexts, which is why we included depth and substratum as explicit factors in our analysis and presentation of patterns of barrens across Tasmania. Our projected barrens cover of up to 50% of eastern Tasmanian reef by 2030s, is possible based on the distribution of reef depths and substrate types across the coast. Sea urchins and sea urchin barrens also relate to different spatial scales (sites and sub-sites) and correlate, either positively or negatively with other reef species (i.e., negative correlation with increasing lobster abundance), supplementary Table 5.

*-It is interesting that in this system ecosystem collapse was due to the poleward expansion of a particular grazer species (*C. rodgersii*) into novel habitat. Perhaps the early warning sign is not just the emergence of incipient barrens, but specifically incipient barrens formed by a particular species in novel habitat. If this is correct, please emphasize this in the manuscript. It is not just range extensions or redistribution of grazers with specific thermal affinities, but barrens have been documented globally and attributed to native grazers as well. The authors mention the inclusion of the native sea urchin as a covariate, but if it is indeed the case that these barrens are driven by the extension of *C.**

rodgersii and not the native species, then this should be emphasized. This could be supported with the addition of density of the native species in Figure 2C or as a supplementary figure.

-Unless I am missing something, why not consider the total absolute density of both the native urchin and also *C. rodgersii*? It looks like they are treated independent in Table S15. It would be interesting to see a plot of the baseline and temporal densities of both species over time.

Author reply:

Native urchins *Heliocidaris erythrogramma* are not observed to form barrens on the open coast of eastern Tasmania, but rather they feed on drift algae along this moderately exposed coastline where macroalgal production is high (Ling et al. 2010; see full reference below). For this reason, total urchin density is inappropriate as a predictor of overgrazing across the study region and thus species need to be kept separate for analyses. To clarify this, we have included this statement in the *Methods* section and added reference to the study which details this phenomenon.

Notably, the phenomenon of poleward propagation of change in spatial patterning is evidently a general phenomenon for reef ecosystems given that herbivory is much higher in tropical systems. Ongoing warming of temperate reefs will likely see increasing impacts of herbivores via the poleward direction with alteration in patchiness of reefs habitats likely driven by the phenomenon of “tropicalization” (see Verges et al. 2014; reference #9 in manuscript).

Reference added: Ling, S.D., Ibbott, S., Sanderson, J.C., 2010. Recovery of canopy-forming macroalgae following removal of the enigmatic grazing sea urchin *Heliocidaris erythrogramma*. *Journal of Experimental Marine Biology and Ecology* **395**, 135-146 (2010).

Line 122 – I recommend removing ‘resistance and resilience’ unless the authors provide some specific definitions for these terms as context, along with carefully defined statistical tests. Briefly, I interpret resistance and resilience as ecosystem-level responses where resistance is the capacity of a system to maintain community structure (or functioning, depending on goals) despite a perturbation, and resilience is both resistance to- and recovery from-perturbation (sensu Bates et al. 2019). Perhaps there is a different interpretation in this system, but please provide clear definitions of these terms if they are retained.

Author reply:

We appreciate this point by Reviewer #2, we have therefore simplified this section by deleting the terms as suggested. The sentence now simply reads: “Thus, we also explore mechanisms of urchin invasion and overgrazing respectively for different abiotic (i.e., reef substratum types/depth) and biotic covariates (i.e., predatory spiny lobsters plus competing native urchins and abalone abundances).”

The ms uses the terms ‘phase-shift’ and ‘stable state’ in a few different places. It seems that ‘phase-shift’ is used to describe the changes here, which I think is correct, but as a reader it would be helpful to see this defined. Dudgeon et al. 2010 define a stable state as a novel configuration that persists after the environmental conditions return to what they were before a perturbation, which is not the

same of a phase shift. I think the terms are used correctly here, but line 109 uses “phase shift” and 115 uses “alternative states.”

Author reply:

Throughout the manuscript and in prior works, I strictly adhere to the terminology used by Scheffer et al as presented in their seminal review of concepts and terminology on phase-shifts in 2001 (Scheffer et al. Catastrophic shifts in ecosystems. *Nature* **413**, 591-596 (2001)). Accordingly, I refer to phase-shift as a transition in ecosystem state and a discontinuous ‘catastrophic’ phase shift as one that results in an “alternative stable state” with hysteresis (Fig. 1c. Scheffer et al. 2001). On line 115 of our manuscript, we explicitly used the term “alternative stable state” as opposed to “alternative state”, the later which doesn’t necessitate that a catastrophic phase-shift has occurred unless the word ‘stable’ is included. We have reviewed our use of terminology and maintain that it is consistent with that defined by Scheffer et al 2001.

Perhaps I missed this, but I don’t see how the critical ‘tipping points’ were derived other than an approximation from Figure 2C. I looked up reference 17 but I don’t see a mention of the tipping points included here. The authors have the data to quantitatively derive these tipping points. The way I have seen this done (and recommend) is that a discriminant analysis is first used to assign each replicate (quadrat or site) a binary value (1 = barren, 0 = forest) based on several predictors (substrate type, algal cover, depth, etc.) and this becomes the response for a logistic regression. The logistic regression is then performed using urchin density as the predictor and the tipping point thresholds are the density values at which it becomes likely (50%, 80%, 95%, whatever probability you choose) that the system will shift to barren or return to forest. The challenge with the recovery arrow is it depends on replicates where barrens were observed transitioning to forests. Without that, it is just a single tipping point based on correlative observations between kelp density and urchin density. Please consider this type of a model to identify the tipping points and whether there are sufficient ‘reverse-shift’ replicates to identify the reverse-shift threshold.

Author reply:

We thank the reviewer for identifying this error in referencing that was not updated within the figure caption during prior edits of the submitted manuscript. The derivation of the tipping points is from reference #4 in the manuscript, not 17. This error has been rectified in the revised manuscript. We agree that dually estimating overgrazing and kelp recovery tipping points is essential and this has been a core research focus. The reverse tipping-point is also estimated in Ref 4.

Minor comments:

I am curious about kelp cover over time. Without a plot of the time series of kelp cover it is difficult to assess whether increases in urchin density proceed kelp loss or just amplify macroalgae declines that are the result of already poor environmental conditions. I am not sure whether these data are available since the study used two discrete time periods, but in order to really attribute barren expansion as a driver of kelp loss (which completely makes sense), it requires a temporal component. Otherwise it is correlative and kelp loss could have preceded increases in urchin density, or led to emergence from refuge.

Author reply:

Increase in barrens cover has been dramatic through time, however outside of the grazed areas caused by incursion of *Centrostephanus*, there has been a slight increase in kelp cover overall. This counter-intuitive phenomenon is the result of a shifting dominance hierarchy of kelp species within Tasmanian kelp beds, whereby there has been major local loss of surface canopy forming giant kelp forests in Tasmania (formed by *Macrocystis pyrifera*, Johnson et al. 2011), and compensatory increase in lower-lying canopy species of *Ecklonia radiata* and *Phyllospora comosa*, that are more tolerant to warming and lower nutrient coastal conditions, with both of these macroalgal species increasing in cover by 13% and 16% respectively (see Supplementary Table 2). This changing of composition in Tasmania kelp beds is the focus of a subsequent manuscript examining phase-shifts in kelp types. It is certainly true to say that in the case of giant kelp that kelp losses have preceded increase in urchin abundance across much of eastern Tasmania, but most of the signal of loss in giant kelp occurred before the coast-wide urchin surveys commenced in 2001/02.

This is clearly stated in the manuscript on ~148-155:

“Notably, the overgrazing collapse of kelp cover was counter to an overall increase in kelp cover outside of reef areas grazed by urchins, with kelp cover on non-grazed reef in eastern Tasmania increasing from 2001/02 to 2016/17 due to compensatory thickening in cover of lower-lying warm-tolerant stipitate species (i.e., *Ecklonia radiata* & *Phyllospora comosa*) in the absence of surface-canopy giant kelp forests (formed by *Macrocystis pyrifera*) which declined (**Table S12**)”

Line 69-72 – this is really interesting that C. rodgersii are so prevalent!

Author reply:

Yes, this species of sea urchin is very prevalent and a key structuring component of reef ecosystems across south-eastern Australia. The huge ecological role played by this urchin across New South Wales represents a serious and ongoing threat, and it is the increasing prevalence of incipient barrens that is sounding this early-warning across the poleward range-extension region.

Line 77 – do lobsters prey on C. rodgersii specifically, or would this be a novel species interaction?

Author reply:

Yes, here we refer to prior description and quantification of lobsters as predators of *C. rodgersii* (i.e., reference #15 - Ling et al PNAS 2009).

Line 127 – It is not just the effect of absolute abundance but also change in grazer behavior. Some highly forested areas can maintain high densities of cryptic passively-grazing sea urchins, so it is the behavioral emergence from refuge resulting from resource availability (decline in macroalgae due to warming) that can feedback to influence declines (Smith & Tinker 2022, Karatayev et al. 2021).

Author reply:

Note that as an obligate scrapping herbivore, foraging behaviour of the sea urchin *Centrostephanus rodgersii* remains surprisingly fixed compared to *Strongylocentrotid* urchins of the northern hemisphere or indeed the native *Heliocidaris erythrogramma* in Tasmania (see reference #24 in the manuscript - Flukes et al. 2012). While low supply of drift algae will shift behavioural switching from benign to active grazing for other urchin species, the obligate scrapping mode of *C. rodgersii* appears largely independent of drift algal supply.

In Figure 2A it looks like the mid latitude sites had historically high densities relative to the north and south (with the exception of St. Helens) and also the relatively highest densities after 2016. Were these deep sites with large boulders? It looks here that the pattern is the same, just amplified. I.e., sites with historically high densities of urchins continued to have the relatively highest densities.

Author reply:

While there is an overall decline in the abundance of *Centrostephanus* across the east coast of Tasmanian from north to south, and there has been greatest increase in urchin abundance and overgrazing on deep boulder reef, the proximity of sites to the warm East Australian Current is the chief driver of local urchin abundance along the east coast (see reference #14 - Ling et al. 2009 GCB). Reviewer #2 is correct in identifying that sites with high urchin abundance in the initial survey period continued to have high urchin abundance in the second survey period, but which had continued to build in abundance through time. To emphasise this feature, we have also pointed to the nil 'Site by Time' interaction effect, i.e., the effect of time on increase in urchin abundance occurred consistently across 'Sites'.

To emphasise these points, the results have been edited to include clarifying statements in parentheses, the relevant section now read:

"However, abundance of the range-extending urchin was uneven at the ~20 km site-scale (chiefly related to the proximity to the East Australian Current¹⁴), a feature which persisted through time (nil site by time interaction effect), and increase in abundance varied significantly at the local ~2 km subsite-scale (Table SI1a)."

- 14 Ling, S., Johnson, C., Ridgway, K., Hobday, A. & Haddon, M. Climate-driven range extension of a sea urchin: inferring future trends by analysis of recent population dynamics. *Global Change Biology* **15**, 719-731 (2009).

Lines 140-142 – the difference in slopes seems to be heavily influenced by two or three quadrats in the 01/02 period with high kelp cover despite high urchin densities (blue points, Figure 2C), but there is clearly less kelp at lower urchin densities in the 2016/17 period. It would be worthwhile to at least compare the slopes after remove those two or three extreme quadrats.

Author reply:

Outlier removal was performed but didn't alter biological conclusions drawn from the relationships. Outlier removal serves to slightly flatten the relationship between kelp bed cover and urchin density, noting that it is the steepening of the relationship in 2016/17, relative to 2001/02, that shows the collapse to barrens around the overgrazing tipping-point at ~ 2.2 urchins m^{-2} (or 700 grams urchins m^{-2}) after Reference #4 - Ling et al. 2015 Phil. B.). Figure with and without apparent outliers is shown immediately below.

Line 289-129 – please parenthetically include the error associated with these means.

Author reply:

Standard errors have now been included and are indicated in parentheses after each stated mean.

Figures:

Figure 1. Love this figure! However “Barrens patch size” is difficult to read. Please make the text black.

Author reply:

Readability of font for inset “Barrens patch size” text has been improved by increasing font size slightly and using a darker red colour. The suggestion to make the font black didn't improve visibility of this title and without being a shade of red the connection to the red inset plot was lost, hence we opted for darker red and larger font-size.

Figure 2. In a) there was clearly an increase in the mean density between periods, but there was an even more pronounced increase in the range across replicate quadrats with a skewness towards a wider density range above the mean. This is an interesting result that could suggest underlying patch-level dynamics. In a and b) please increase the transparency of the filled area and show the datapoints. c) the reverse shift arrow is difficult to see. Please change the color or make it darker.

Also, I know that the tipping point arrows are approximate, but where these approximated by visual inspection or using a quantitative approach? Unless I am missing something, I don't see how the 2.2 and 0.36 thresholds were determined. Please see my comment about the discriminant analysis and consider whether the recovery arrow is appropriate depending on the number of replicates where true recovery was observed. If recovery was not observed, then hysteresis is speculative and the relationship between kelp cover and urchin density should be represented by a single line or curve (single tipping point for forward and reverse shifts).

Author reply:

This is a box whisker plot whereby the data is summarised by showing median value plus the upper and lower quartiles of the data indicated by upper lower edges of box respectively, whiskers indicate the upper (25%) and lower (25%) extents of the data, with dots representing outliers beyond these extents. As such, altering transparency will not show additional data points as suggested. All data could be shown; however, the box whisker data summaries are much easier to interpret given the clouds of data across 13 sites surveyed in 2 time periods each. Therefore, we have not altered the box whisker plot.

Reverse shift arrow has been edited to increase visibility and is now same darker shade as per forward shift arrow.

Figure 3. Please add the datapoints and make the filled area more transparent so that we can see them. I would reorder from bottom to top: flat rock, small boulders, large boulders.

Author reply:

Adding of individual data points was attempted but maintaining figures as box whisker summary plots of the data were much cleaner and easier to interpret. We have reordered the panels as requested, i.e., panels have been re-ordered from large boulders to small boulders to flat rock which better aligns with description of results in text. Thanks to the Reviewer #2 for picking this up.

Figure 4. Please add text indicating the patch size to each facet to remind us the extent of 'small' 'medium' and 'large'.

Author reply:

As requested, patch sizes for each incipient barrens patch type have been added to the Figure 4 caption.

References in response:

Filbee-Dexter, K., & Scheibling, R. E. (2014). Sea urchin barrens as alternative stable states of collapsed kelp ecosystems. Marine ecology progress series, 495, 1-25.

Karatayev, V. A., Baskett, M. L., Kushner, D. J., Shears, N. T., Caselle, J. E., & Boettiger, C. (2021). Grazer behaviour can regulate large-scale patterning of community states. *Ecology Letters*, 24(9), 1917-1929.

Smith, J. G., & Tinker, M. T. (2022). Alternations in the foraging behaviour of a primary consumer drive patch transition dynamics in a temperate rocky reef ecosystem. *Ecology Letters*, 25(8), 1827-1838.

Bates, A. E., Cooke, R. S., Duncan, M. I., Edgar, G. J., Bruno, J. F., Benedetti-Cecchi, L., ... & Stuart-Smith, R. D. (2019). Climate resilience in marine protected areas and the 'Protection Paradox'. *Biological Conservation*, 236, 305-314.

Reviewer #3 (Remarks to the Author):

Summary:

Shifting environmental conditions are redistributing species across the globe. This study focuses on a range expansion of a sea urchin native to Australia to kelp forests off the coast of eastern Tasmania. Replicate surveys conducted across 13 sites and 15 years apart are described. These include diver surveys of urchin, lobster, and abalone abundance, and percent cover of substrate, including barren. Additionally, the authors conducted similar video surveys of planar cover of total urchin barrens and of different barren sizes in order to show the evolution of small incipient barrens (a starting condition) to large and continuous barrens that represent a state shift in the system. The authors found that incipient barrens increased through time, especially in northeastern and eastern Tasmania (southeastern Tasmania showed few urchins and barrens). The authors also showed how increases in the abundance in urchins was associated with disproportionately increased barren area. Deeper areas of reef with large boulders were associated with greater urchin abundance and there were negative associations between urchin abundance and the predatory lobster. The authors conclude with a discussion of using these early warning signals to initiate further research, partnerships, and ecosystem defense strategies.

Author reply:

No reply comments or changes to the manuscript in reply to these summary comments by Reviewer #3, other than to say Reviewer #3 presents an accurate summary and clear comprehension of some of the key features of the research. We thank Reviewer #3 for their time and effort in reviewing and helping us clarify our manuscript where identified.

General comments:

Overall, I thought this was a well written and timely study worthy of publication. The two most important results from the study, in my opinion, are the data showing the temporal progression of urchin abundance and barren cover/size across space and the clear change in slope between barren area and urchin density between the two time points, implying a non-linear relationship. I did have some comments on how to improve the manuscript and reproducibility.

Author reply:

We thank Reviewer #3 for their encouraging summary and attention to detail in highlighting suggested improvements, which are addressed in full below and within the revised manuscript.

First, I have some trouble with the authors stating that barren cover may increase to 50% by 2029. I understand that this is an extrapolation on the measured barren expansion rate between the two survey dates. I also feel that the authors intuition is likely correct, especially considering that there are multiple stressors that are all trending towards a more hostile environment for kelp. However, I am not sure the data support this conclusion when viewed in a different way. First, we start with looking at sites 1 – 9, defined as eastern Tasmania by the authors. From the planar barren data in Supp Data 3, the site with a highest urchin barren area in the 2001-2002 surveys (Site 2; St Helens) had a mean urchin barren area of ~23%. In the later surveys in 2016-2017, it had a mean urchin barren area of ~42.5%, representing an increase of just under 20% in 15 years. All of the other sites had vastly lower barren areas in the 2001-2002 surveys, and sites 1, 3, 4, 8, and 9 all has less than 10% barren area in the 2016-2017 surveys (with site 5 having just over 10% and site 6 and 7 having about 17.5%). It could be that there are non-linearities in the rate of barren expansion that are not being considered when observing the early stages of barren formation (i.e., is there a greater rate of barren expansion when moving from 0% to 10% than from 20% to 50%) at sites using only two time points. While I personally believe the authors may be correct (due to both increases in urchin abundance and temperature in this system, along with personal experience observing urchin barrens in other areas of the world), I would caution the authors to word these statements carefully.

Author reply:

We appreciate the cautionary comments from Reviewer #3 regarding projections of barrens cover by 2029. This calculation is based on the compounding rate of increase per year interpolated from the change observed over the 15-year period, averaged across sites 1-9. It was performed to provide a gauge as to the potential scale and immediacy of the threat of overgrazing based on the monitoring data at hand. As such we feel there is value in retaining this calculation and note that as Reviewer #3 suggests, there appears to be non-linearities at play in the urchin population increase and extent of overgrazing along the coastline. That is, in contrast to the relatively stable urchin densities within the northeast at the Site 2 St. Helens, where most available reef substrate (particularly deep boulder habitat) has been already overgrazed, the urchin population elsewhere has shown greater relative change consistent with the notion of an exploding population dynamic.

Notably, beyond the data presented in the current manuscript, sampling of reefs in the south-east of Tasmania in 2021/22 (site 9) has revealed a doubling of urchin densities within the past 5-years, whereas the population previously took 15-years to double from (2001/02 to 2016/17); equating to a tripling in population increase in the most recent years. Intensive harvesting and culling activities have been directed to this area to mitigate the explosion of urchins. Follow-up modelling research, using a combination of fisheries catch data and changes in urchin densities on the reef, is partitioning the effectiveness of urchin harvest in mitigating the accelerating population increase and rapid increase of incipient barrens formation (e.g., Fig. 4a, site 9).

We appreciate the broad nature of our projection, and note the cautionary words of Reviewer #3, and have therefore further qualified the statement in question, which now

reads: "Increase in barrens equated to an expansion rate of 10.5% pa, with projection of this unmitigated rate indicating barrens cover is poised to increase to ~50% in eastern Tasmania (sites 1-9) as soon as the year 2029, consistent with that observed locally at St. Helens (site 2) and as observed in NSW¹⁶."

Additionally, while the authors do state that eastern Tasmania only covers sites 1 – 9 (and not the southern sites 10 – 13 where little barren cover has been observed in the data presented), there is a chance that people will interpret these findings as the entire eastern coast of Tasmania. Another reason to be as clear as possible when stating these predictions in the manuscript.

Author reply:

We thank Reviewer #3 for this sage advice also. To this end, we've added explicit mention of sites 1-9 to the statement of projection of barrens expansion rate (see reply to query immediately above).

My other concern is with the data files that accompany the manuscript. I found a few discrepancies (or edits) that need to be addressed before publication. First, in Figure 2, n is defined as 1730 for the 2001-2002 surveys and 1727 for the 2016-2017 surveys, but the data file contains 1600 for both. I see the data files says (in the sheet name) that 'no sand' has been truncated. This may lead to the discrepancy and must be addressed. I found the stated statistics in the figure to be nearly identical.

Author reply:

We thank Reviewer #3 for picking up on this discrepancy. Yes, the difference in number of quadrats was due to the removal of quadrats dominated by sand in the dataset "Data SI3". The figure has been replotted for consistency with n=1600 for both survey periods, with trends in each period remaining largely unchanged following removal of 130 and 127 sand dominated quadrats from 2001/02 and 2016/17 respectively. Thanks again to Reviewer #3 for highlighting this discrepancy.

Second, using the planar urchin barren cover I found sites 1 – 9 to have 3.22% barren cover in the 2001-2002 surveys and 12.72% in the 2016-2017 surveys. Similar to what is stated in the paper but different (it may also lead to a slightly different rate of change for your extrapolation). Maybe I am looking at the wrong data, but others may do the same.

Author reply:

The experimental design used to present and analyse the data is a fully hierarchical spatial design with site means calculated from mean of 3 sub-sites per site, with sub-site means calculated from mean of 4 transects per sub-site, within each time period. Estimates of urchin density or percentage cover of barrens therefore differ slightly depending on whether all data are averaged versus the correct averaging within each spatially hierarchical level of the design. Reviewer #3 calculates means within each period by pooling across all spatial scales of the sampling design, as opposed to calculating means within a particular spatial scale before estimating variability at the level above. To supplement the spatial hierarchical

design as detailed in the Methods, we now explicitly state how the means for urchins and barrens percentages are calculated in the results to make this clearer.

That is, lines ~134-138 in the Results now reads: “Across the region of range-extension (sites 1-13), dive surveys to 18 m depth revealed the range-extending urchins’ abundance to increase significantly from an average density of 0.10 (± 0.05 SE; n=13 sites) to 0.18 (± 0.06 SE, n=13 sites) individ. m⁻² from 2001/02 to 2016/17 (Fig. 2a; Table S11; with site means calculated hierarchically from the mean of 3 sub-sites per site, with sub-site means calculated from mean of 4 transects per sub-site within each time period)”.

Third, I was a bit confused on the proportion of barren sizes in the planar barren area data sheet. I think these data need more explanation in the manuscript so that readers can analyze these data correctly.

Author reply:

We have clarified the calculations of the planar barren cover percentage in the Methods section. The data header for the overall planar barrens cover data has also been extended to explain how the calculation is made from the 4 barrens types. The header now reads: “Overall_planar_percent_barrens (calculated by summing all barrens types after multiplying each type by mid-point proportions of 0.925, 0.625, 0.30 and 0.10 for continuous barrens and large, medium and small incipient patches respectively)”.

These are the three that were most apparent in my analysis of the data and there may be others, so I think a careful reanalysis of the data provided is warranted.

Author reply:

We thank Reviewer #3 for their detailed exploration of our figures and associated data files. As detailed above, we have reviewed and trimmed the data in Fig. 2c by removing quadrats dominated by sand such that the dataset is all square with that presented in the accompanying data file. As such, we have now achieved consistency for this dataset throughout the manuscript. We have also reiterated the hierarchical spatial design within the results which now better supplements that detailed in the Methods section.

Specific comments:

Define the abbreviation for New South Wales (NSW) in paragraph three of the introduction.

Author reply:

Fixed. Thanks to Reviewer #3.

Misspelling of ‘survey’ in the acknowledgments.

Author reply:

Fixed. Thanks to Reviewer #3.

Reviewer #4 (Remarks to the Author):

This paper shows an extensive amount of kelp and sea urchin data collected in Tasmania, comparing 2001/2002 and 2016/2017. The authors stress the current increases in urchin barren cover in warming coastal zones (poleward). Besides, they focus on the potential for early-warnings of a catastrophic shift towards complete urchin barrens, by monitoring incipient spatial patterns.

As a theoretical ecologist, I'm not familiar with the field methods used, so I won't be able to comment on these. I will focus on the link to the theory made by the authors, and the interpretation on the data in the context of alternative stable states and early warning.

First of all, I'm impressed by the richness of the data. I'm not completely convinced of the entire line of reasoning, and the support by the data. I would like to share my thoughts on this, and some suggestions below.

Firstly, I miss a discussion on the scale of the theoretical dynamics and the data. Are the observed patterns assumed to be stable, or unstable transient states where small incipient patches with high numbers of urchins slowly expand into the kelp? In the last case, is the early warning not actually the start of the (maybe slow) collapse? Or in the first case, how catastrophic is the collapse then? A reference is made to Rietkerk et al, but any discussion on the scale of the described feedbacks, the scale of the tipping points, and the scale of the data is lacking.

Author reply:

We thank Reviewer #4 for helping us to better link theory to our empirical data. The observed pattern of increasing incipient grazed barrens patches by the invading urchin is most certainly currently an unstable transient state as the urchin population undergoes population explosion. We use the term 'incipient' to mean: "beginning to happen or develop". These barrens patches are becoming more numerous and expanding into the kelp, with observed coalescence of incipient patches leading to local kelp collapse. To help clarify this, a statement to this effect has now been added to the manuscript at line 110-113:

"Incipient barrens represent a transient state progressing from small grazed-patches (1 m² scale) that become more numerous and coalesce to ultimately collapse kelp and form extensive barrens (10,000 - 1,000,000 m² scale) as urchin abundance locally increases (**Fig. 1a-d**)."

The scale of the study is indicated by the spatially hierarchical sampling design, sampled 15 years apart, which identifies the urchin population increase and increasing presence of incipient barrens patches across the eastern Tasmanian region being invaded by the urchin. The spatial scales of the whole eastern Tasmania coast (~250 km), sites within this coast (13 sites spaced every ~20 km along the coast), sub-sites within sites (spaced every ~2 km), and individual transects (spaced every ~200 m), with base scale of 5 m² quadrats occurring within each transect, is made explicit in the Methods. To clarify this, the spatially hierarchical nature of this experimental design is now also reiterated within the first mention of results to clarify scales of investigation and ultimately the observed collapse, and early warning signs of pending collapse, across the coast. The scale of urchin barrens is shown in the snapshot of barrens patch-size distribution across the Tasmanian east coast is shown in Fig. 1.

To further clarify the transitory dynamics of the incipient barrens, the Fig. 1c&d caption now reads: “(c) Schematic of the spatial pattern of ecosystem collapse from kelp forests to urchin barrens via intermediate ‘incipient barrens’, which expand and coalesce as urchin density increases across reefs. (d) Schematic of observed climate-driven spatial pattern formation in the poleward direction; following from bottom to top provides a space-for-time substitution of the process of collapse from kelp beds to extensive barrens via appearance and ultimately coalescence of incipient barrens as the overgrazing tipping-point in urchin density is increasingly exceeded.”

One of the first points made in the paper, is that the kelp-sea urchin system has alternative stable states, and shows catastrophic shifts at a certain level of urchin cover. This data seems to support this, although it is not completely clear to me at what scale (see first point) (if I understand correctly 50m). To illustrate the hypothesized existence of alternative stable states, it would be insightful to illustrate the frequency distribution vertically next to Fig. 2c, to show bimodality in the data.

Author reply:

As stated in the caption for Fig. 2c, the scale of quadrats is 5 m². As suggested by Reviewer #4, we have illustrated the existence of alternative stable states by now including a frequency distribution (arranged vertically) on the right-hand side of Fig. 2c, which shows increasing bimodality through time. We thank Reviewer #4 for this suggestion which now better links the theory of alternative stable states to the overgrazing collapse of kelp beds that is underway.

The authors draw two arrows in Figure 2c indicating tipping points. I understand that these levels are based on earlier research, but since they have no relation to the data studied itself, the drawn trajectories give the reader a bit of a false idea of a directional change. I would suggest to draw actual observed trajectories by drawing arrows for each of the sites between the two observation periods.

Author reply:

This is correct, the drawn lines represent the contrasting forward (overgrazing) and reverse (kelp recovery) tipping-points derived from prior study (manuscript reference #4 – Ling et al. 2015). The importance of this figure is to show the collapsing (steepening) relationship between kelp and urchin density, which is consistent with the prior cited research and concepts of collapse, i.e., catastrophic phase shift (after Scheffer et al 2001 highlighted above in reply to Reviewer #2). While we appreciate the suggestion of drawing trajectories between every blue and red dot to show individual transitions through time, the quadrat data (averaged at 5 m²) are not perfectly fixed in space between time periods, hence while we can show the overall change in relationship, individual quadrats may not represent the exact same space. We have therefore not attempted to link each and every quadrat.

It is unclear to me how Figure 3 relates to the main point of the paper. Maybe I miss something, and could it just be stressed a bit more.

Author reply:

Figure 3 provides important natural history information about the depths and reef substrate types where the widespread collapse to barrens has occurred, or is impending, across eastern Tasmania. That is, the greatest increase of urchin population and overgrazing impacts greatest in deeper water on boulder dominated reefs. Without this practical information, applied efforts to mitigate kelp ecosystem collapse would be misguided.

We stress this point on lines ~128-132, where we state: “If early-warning signals are ultimately detectable, then knowing how to act, before it is too late, is contingent on understanding mitigating ecological circumstances. Thus, we also explore mechanisms of urchin invasion and overgrazing respectively for different abiotic (i.e., reef substratum types/depth) and biotic covariates (i.e., predatory spiny lobsters plus competing native urchins and abalone abundances).”

How the data is presented at the moment, I don't see that mapping patches and patch sizes would help for early warning. How do the patch sizes compare to urchin barren cover? Can results from Figure 2 and patch sizes as measured in Figure 4 be compared?

Can actually urchin barren cover be used as an early warning sign, where simply the occurrence of patches is the first warning (so from 0 to >0% cover), or is there more information in the patch sizes measured? For instance, does cover stay the same, but patches become larger at sites where there are more urchins? This could easily be shown in a plot, and helps the reader to convince the usefulness of looking at patches and patch sizes in particular.

Author reply:

A barrens patch is defined as a section of reef where all the kelp within the patch has been grazed away by urchins resulting in bare-rock or encrusting/ filamentous algal forms (as seen in images of barrens patches within the manuscript, e.g., Fig. 4a&b insets). So, the percentage cover of kelp within a barrens patch is 0%, conversely the percentage cover of barrens within a barrens patch is 100%. That is, the cover of kelp (0%) or barrens (100%) within a barrens patch stays the same, but patches become larger at sites where there are more urchins and neighbouring patches start coalescing to form larger scale features.

The Methods section has been edited to clarify this, and includes citation of prior research defining this relationship (i.e. Flukes et al. 2012 – Reference #24 in the manuscript):

“Note that a barrens patch is an area of reef where all the kelp within the patch has been completely grazed away by urchins, resulting in bare-rock or encrusting/ filamentous algal forms (e.g., Fig. 4a&b insets), i.e., the percentage cover of kelp within a barrens patch is 0%, or alternatively 100% barren cover. As barrens patches become larger, urchin abundance also increases²⁴.”

Overall, I think the link between theory and data can be strengthened more to convince the reader. Also, the bridge to actual early warning signs (cover? patches?) needs some consideration.

Furthermore, I think with some adaptations, it can become a rich paper, discussing an important question, whether (the increase of / occurrence of) incipient patches can be used as warning signs of systemic kelp collapse.

Author reply:

The actual early warning sign is the appearance of many small incipient barrens patches that can coalesce to larger features, which become self-maintaining and are very hard to reverse once the tipping-point of overgrazing is exceeded across increasingly large scales (see Reference #4 in the manuscript, i.e., Ling et al. 2015). As we show in our long-term study, it is the appearance of numerous small patches that is the early warning sign. To stress this point, we have now added a concluding statement explaining that tracking the appearance of these patches across space using remote monitoring methods (such as towed underwater video, independent of labour-intensive and spatially restricted dive surveys) is a rapid way of mapping this worsening problem and identifying where tactical efforts to mitigate overgrazing collapse must be directed before it is too late. The following statement has been added to the discussion (line 217-220): “Notably, in comparison to spatially restrictive and depth-limited dive surveys, towed-video proved an efficient scalable and cost-effective monitoring method well-suited to high-frequency spatially extensive sampling, which are key considerations for maximising detection of spatial-pattern-formations indicative of impending ecosystem change.”

To further help bridge the theory to practice, we also added additional words in parentheses within the concluding paragraph. The fourth last sentence in the conclusion now reads: “Beyond previous studies, we reveal that early-warning of ecosystem collapse under climate change can be recognized well-in-advance by looking equatorward for signs of what to expect (i.e., looking in the equatorward direction to detect emergence of novel spatial pattern formations, see **Fig. 1d**).”

We thank Reviewer #4 for their encouraging words and for highlighting opportunities for improving the link between theory and our data.

REVIEWERS' COMMENTS

Reviewer #2 (Remarks to the Author):

The revised manuscript is well written and will be an important contribution to the literature. The authors thoroughly addressed my major concerns in the revised manuscript. I recommend the article for publication after the authorship team addresses the following:

Line 27 – I believe 50% deforestation by 2030 is possible, but I still feel that this statement is somewhat speculative. If this statement is retained in the abstract, I recommend the authors qualify it similarly to the text in the body where it appears (e.g., the spatial scale of the projected 50% loss of forests is specific to eastern Tasmania).

Thank you for mentioning in the comments that *C. rogersii* is an obligate scraping herbivore. This is a very important trait that contrasts the feeding behavior of the native sea urchin. As a reader unfamiliar with this system, it would be helpful to know early in the paper this important differentiation, especially since northern hemisphere species can switch between foraging on detritus vs. grazing on live macroalgae.

I have a related question about the foraging behavior of *C. rogersii* that may warrant some further clarification in the main text. Since this species is an obligate scraping herbivore, what is the primary mechanism of deforestation of macroalgae? Is it that *C. rogersii* consumes the holdfasts of adult kelps, thereby directly contributing to deforestation? Or do they consume gametophytes or newly recruited kelp and therefore adult kelps are unable to be replenished?

Lines 178-192 – please report summary statistics where ‘significantly’ is used.

Reviewer #3 (Remarks to the Author):

Thank you for addressing my comments in a clear manner. I would also like to thank you for making the methods used more clear to the reader and addressing the issues with the underlying data files.

REVIEWERS' COMMENTS

Reviewer #2 (Remarks to the Author):

The revised manuscript is well written and will be an important contribution to the literature. The authors thoroughly addressed my major concerns in the revised manuscript. I recommend the article for publication after the authorship team addresses the following:

Line 27 – I believe 50% deforestation by 2030 is possible, but I still feel that this statement is somewhat speculative. If this statement is retained in the abstract, I recommend the authors qualify it similarly to the text in the body where it appears (e.g., the spatial scale of the projected 50% loss of forests is specific to eastern Tasmania).

Author reply:

We thank Reviewer #2 for their comments. This statement has been modified in the abstract as suggested. The sentence now includes more definitive mention of the relevant spatial scale to now read:

“Demonstrating poleward progression of collapse over 15-years, early warning ‘incipient barrens’ are now widespread along 500 km of coast with projections indicating that half of all kelp beds within this range-extension region will collapse by ~2030.”

Thank you for mentioning in the comments that *C. rodgersii* is an obligate scraping herbivore. This is a very important trait that contrasts the feeding behavior of the native sea urchin. As a reader unfamiliar with this system, it would be helpful to know early in the paper this important differentiation, especially since northern hemisphere species can switch between foraging on detritus vs. grazing on live macroalgae.

I have a related question about the foraging behavior of *C. rodgersii* that may warrant some further clarification in the main text. Since this species is an obligate scraping herbivore, what is the primary mechanism of deforestation of macroalgae? Is it that *C. rodgersii* consumes the holdfasts of adult kelps, thereby directly contributing to deforestation? Or do they consume gametophytes or newly recruited kelp and therefore adult kelps are unable to be replenished?

Author reply:

We have now clarified the feeding mode early on in the manuscript. Sentence in the Introduction edited to read:

“Within the central and southern NSW native range of *C. rodgersii*, this urchin via its obligate scraping mode of feeding on macroalgae including consuming holdfasts and juvenile macroalgal recruits, maintains barrens over more than 50% of all shallow reef 16 and similar levels of overgrazing are now apparent at some sites in northeast Tasmania.”

Lines 178-192 – please report summary statistics where ‘significantly’ is used.

Author reply:

All summary statistics are contained within the Supplementary Information contained within Table SI5. This is now explicitly stated in the text.

Reviewer #3 (Remarks to the Author):

Thank you for addressing my comments in a clear manner. I would also like yo thank you for making the methods used more clear to the reader and addressing the issues with the underlying data files.

Author reply:

We thank Reviewer #3 for the opportunity to clarify our underlying data. No further comment.